# Efficient population modification gene-drive rescue system in the malaria mosquito *Anopheles stephensi*

Adriana Adolfi [1,10], Valentino M. Gantz [2], Nijole Jasinskiene[1], Hsu-Feng Lee [1], Kristy Hwang [1], Gerard Terradas [2,3], Emily A. Bulger [2,3,11,12], Arunachalam Ramaiah [4,5], Jared B. Bennett[6], J. J. Emerson [4], John M. Marshall [7,8], Ethan Bier[2,3] & Anthony A. James [1,9✉]

Cas9/gRNA-mediated gene-drive systems have advanced development of genetic technologies for controlling vector-borne pathogen transmission. These technologies include population suppression approaches, genetic analogs of insecticidal techniques that reduce the number of insect vectors, and population modification (replacement/alteration) approaches, which interfere with competence to transmit pathogens. Here, we develop a recoded gene-drive rescue system for population modification of the malaria vector, *Anopheles stephensi*, that relieves the load in females caused by integration of the drive into the *kynurenine hydroxylase* gene by rescuing its function. Non-functional resistant alleles are eliminated via a dominantly-acting maternal effect combined with slower-acting standard negative selection, and rare functional resistant alleles do not prevent drive invasion. Small cage trials show that single releases of gene-drive males robustly result in efficient population modification with ≥95% of mosquitoes carrying the drive within 5-11 generations over a range of initial release ratios.

[1] Department of Microbiology & Molecular Genetics, University of California, Irvine, CA 92697-3900, USA. [2] Section of Cell and Developmental Biology, University of California, San Diego, La Jolla, CA 92093-0349, USA. [3] Tata Institute for Genetics and Society (TIGS)-UCSD, La Jolla, CA 92093-0335, USA. [4] Department of Ecology and Evolutionary Biology, University of California, Irvine, CA 92697-2525, USA. [5] Tata Institute for Genetics and Society (TIGS)-India, Bangalore, KA 560065, India. [6] Biophysics Graduate Group, Division of Biological Sciences, College of Letters and Science, University of California, Berkeley, CA 94720, USA. [7] Division of Epidemiology & Biostatistics, School of Public Health, University of California, Berkeley, CA 94720, USA. [8] Innovative Genomics Institute, Berkeley, CA 94720, USA. [9] Department of Molecular Biology & Biochemistry, University of California, Irvine, CA 92697-4025, USA. [10] Present address: Liverpool School of Tropical Medicine, Vector Biology Department, L3 5QA Liverpool, UK. [11] Present address: Developmental and Stem Cell Biology Graduate Program, University of California, San Francisco, CA 94158, USA. [12] Present address: The Gladstone Institutes, San Francisco, CA 94158, USA. ✉email: aajames@uci.edu

Challenges faced by current programs to eliminate malaria from high-endemic areas[1] have fostered the development of novel control strategies including those based on genetically modified mosquitoes. These genetic approaches are supported by the development of Cas9/guide RNA- (Cas9/gRNA) based gene-drives[2–4], with pioneering studies demonstrating effective mosquito population suppression[5–7] and modification[8–10], the latter aimed at impairing the ability of adult females to transmit the *Plasmodium* parasites causing the disease.

One population modification approach is to express multiple anti-parasite effector molecules in the form of single-chain antibodies directed against sexual stages of *Plasmodium falciparum* in relevant mosquito tissues[8,11,12]. This strategy relies on efficient copying of a drive system from one chromosome to its homolog in the germline by homology-directed repair (HDR) following the induction of double-stranded DNA breaks by the Cas9 endonuclease. This efficient repair process results in highly biased transmission of the drive element and the rapid spread of the parasite-refractory traits throughout a naive population. However, competing end-joining (EJ) repair mechanisms can result in insertions or deletions (INDELs) that may be resistant to subsequent Cas9-mediated cleavage and may impede the copying process[9,13]. These resistant alleles can be generated at different developmental stages and tissues, including the germline and somatic tissues during early embryogenesis, due to parental deposition of Cas9/gRNA complexes or somatic expression from "leaky" promoters[8,9,14–16]. INDELs at the cut site of a coding region of a gene often disrupt subsequent protein function (nonfunctional EJ), but mutations that maintain protein function (functional EJ) also can occur. Both outcomes can affect the introduction dynamics of the drive element, and in the absence of strategies that prevent or mitigate EJ events[16–18], gene-drives can cease to spread and be eliminated from a population if they carry a fitness cost[9,13].

One resistance-mitigation strategy for population modification approaches is to design a drive that targets an essential gene while concomitantly providing a recoded rescue sequence[19]. Under these conditions, EJ resistant alleles that lack the rescue sequence are eliminated from the population as the result of the fitness load associated with the loss-of-function of the targeted gene. The feasibility of "rescue systems", such as Cleave and Rescue (CleaveR) and toxin-antidote, was demonstrated in the fruit fly, *Drosophila melanogaster*, by targeting essential genes using alternative combinations of *trans-* or *cis-*acting drive elements and rescue sequences[20–22]. These systems are independent of HDR copying of the drive and result in threshold-dependent drive invasion dynamics[20,21]. When arranged in a configuration where the Cas9 and gRNAs are not located in the same transgenic construct, these systems are predicted to remain confined locally[21,22]. Alternative designs based on HDR-mediated spread of rescue sequences display non-confinable invasion dynamics in model-based simulations and *D. melanogaster* cage experiments[23,24].

Here, we report an HDR-based autonomous gene-drive rescue system that eliminates nonfunctional EJ events in *Anopheles* mosquitoes and provide experimental evidence of sustainable long-term population modification in cages. This efficient performance is based on a phenomenon referred to as lethal/sterile mosaicism, recently reported in *D. melanogaster*[25], in which recessive nonfunctional resistant alleles act as dominant deleterious mutations and are eliminated from the mosquito population as they arise. The drive system, designated "Recoded" (Rec), was developed in the Indo-Pakistan malaria mosquito, *Anopheles stephensi*, the major vector in urban areas of India that also has recently invaded the Horn of Africa[26,27]. The drive system is inserted into the autosomal gene, *kynurenine hydroxylase (kh)*,

involved in the tryptophan metabolism pathway[28], and carries a partial recoded *kh* sequence that restores full activity of the *kh* gene (Rec*kh*). Homozygous loss-of-function mutations in *An. stephensi kh* result in a pleiotropic phenotype that includes loss of the black eye pigment (white eyes) and reduced female survival, fertility, and fecundity following blood feeding[8,9]. The recoded *kh* sequence carried by the drive construct supports normal survival and reproductive capacity in females, while females failing to inherit the Rec*kh* construct from their mothers and carrying nonfunctional mutated copies of *kh* are culled from the population. This elimination does not rely solely on the standard negative selection of load-conferring recessive alleles but is driven by lethal/sterile mosaicism resulting from maternally deposited Cas9/gRNA complexes mutating the wild-type (WT) paternal allele. If this lethal/sterile mosaic process occurs in a sufficient number of cells, it can act dominantly to eliminate females that inherit nonfunctional EJ alleles from the reproductive pool. The combination of lethal/sterile mosaicism and negative selection results in robust drive outcomes demonstrable in population cage experiments with initial Rec*kh*-to-WT male seeding ratios of 1:1, 1:3, and 1:9. We also report a Cas9/gRNA-mediated cassette exchange[29] in mosquitoes called "Swap" that permits the rapid and flexible replacement of specific sequences within genomically integrated transgenes without the need for docking sites.

## Results

**Generation of Rec*kh* gene-drive mosquitoes by Swap.** The first *An. stephensi* Cas9/gRNA-based gene-drive system, AsMCRkh2 (referred hereon as "non-recoded", nRec), was inserted into the coding region of the *kh* locus[8]. Homozygous ($kh^{nRec}/kh^{nRec}$) or heteroallelic ($kh^{nRec}/kh^{-}$) loss-of-function *kh* allelic combinations cause a white-eye phenotype and impairment of blood-fed female survival and reproduction, resulting in a suppression drive that limits its spread in caged populations[9].

We modified this loss-of-function prototype drive system by inserting a recoded portion of the *kh* complementary DNA (cDNA) precisely at the 3′-end junction of the gRNA cleavage site producing an in-frame chimeric functional *kh* gene that restores endogenous gene activity ($kh^{Rec+}$) (Fig. 1a). Swap facilitated integration of the recoded cDNA through the coordinated action of two gRNAs in the presence of a donor template carrying homology arms matching the flanking regions of each of the cut sites. A mixture comprising a donor plasmid marked with green fluorescent protein (GFP) (pRec*kh*) and two plasmids, each encoding one of the two gRNAs, was injected into 504 embryos of the DsRed-marked nRec gene-drive line, which carries *vasa*-Cas9 and U6Akh2-gRNA transgenes targeting the *kh* locus in the germline[8]. Successful cassette replacement was visualized by loss of the DsRed marker and concomitant acquisition of the GFP-marked Rec*kh* cassette. Two independent transformation events were recovered from the 184 $G_0$ adults surviving microinjection, one in a female and one in a male founder line, totaling 96 transformants among the 25,293 $G_1$ larvae screened (Supplementary Table 1).

The recoded *kh* sequence carried by the Rec*kh* cassette is integrated at the junction where insertion of the original nRec line disrupted the *kh* gene and restores the coding sequence and gene function (Supplementary Fig. 1). As a result, individuals from these GFP-marked mosquito lines exhibit WT eye color while retaining the core nRec autonomous gene-drive components (Fig. 1b). Therefore, Rec*kh* should sustain efficient copying onto unmodified WT *kh* alleles ($kh^{+}$) while producing a fully functional *kh* allele ($kh^{Rec+}$) encoding a protein product identical to that produced by the endogenous WT locus (Fig. 1c).

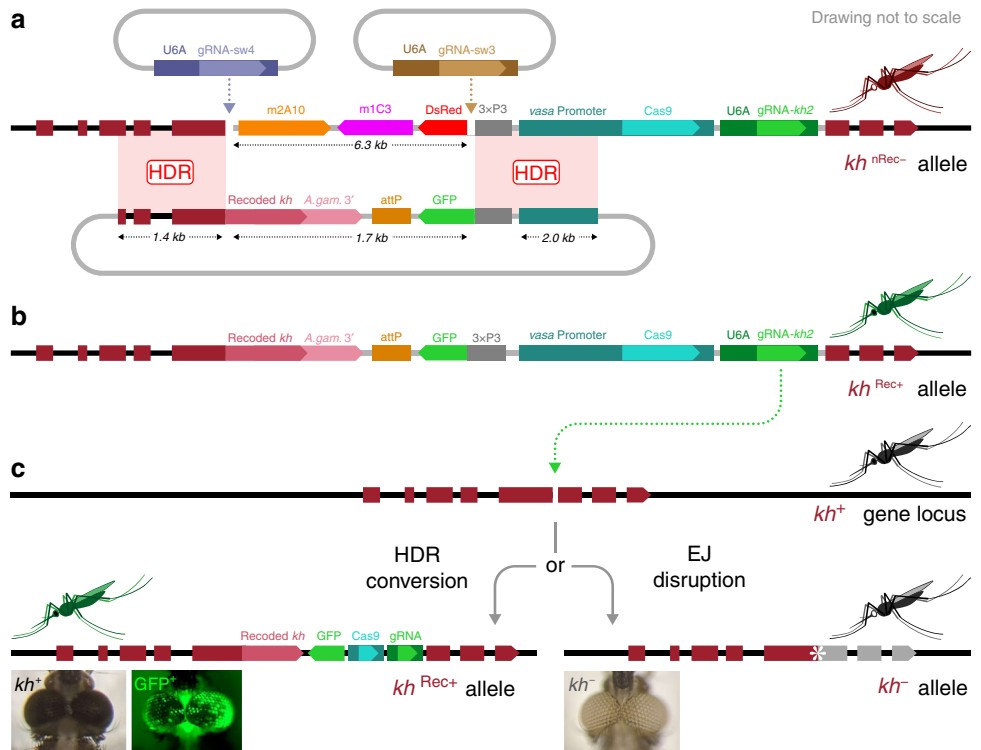

**Fig. 1 The Reckh gene drive. a** Swap strategy for Cas9/gRNA-mediated cassette exchange. Two plasmid-encoded gRNAs (top) guide cleavage in the genome of the white-eyed nRec mosquito line ($kh^{nRec-}$)[8] (middle), leading to the excision of a fragment including the DsRed eye (3xP3) marker and the two antimalarial effectors m2A10 and m1C3[11]. The HDR template plasmid (bottom) carries homology arms flanking either cut site, promoting the insertion of a GFP-marked donor template that carries a recoded portion of the *kh* gene followed by the 3′-end sequence of the *An. gambiae kh* gene including the 3′UTR (A.gam.3′) to minimize homology. **b** The insertion of this unit restores *kh* gene function while creating a sequence ($kh^{Rec+}$) that is uncleavable by the endogenous drive components. **c** The Reckh gene-drive includes an *An. stephensi* codon-optimized Cas9 driven by the germline-specific *vasa* promoter from *An. stephensi* and a gRNA (gRNA-kh2) directed to the fifth exon of the unmodified $kh^+$ gene (top) regulated by the ubiquitous promoter of the *An. stephensi* U6A gene[8]. The cut in the *kh* gene of the Reckh mosquito germline can be repaired by drive integration via HDR (homology-directed repair) or by the less desirable EJ (end-joining) pathway (bottom). HDR results in the integration of the drive cassette that maintains *kh* gene function at the integration site ($kh^{Rec+}$), while EJ usually causes the formation of loss-of-function alleles ($kh^-$). When function is lost in both copies of the gene, individuals with white eyes are produced. *kh*, *kynurenine hydroxylase* gene; attP, *φC31* recombination site; U6A, RNA polymerase-III promoter; gRNA, guide RNAs; Cas9, Cas9 open reading frame; vasa, vasa promoter; 3xP3, eye-marker promoter; GFP, green fluorescent protein; dominant marker gene. The horizontal dimension of the mosquito heads at the eyes in the images is ~1 mm.

**Primary transmission of the Re*kh* element is reduced by maternal deposition of Cas9/gRNA complexes.** Drive performance of the Re*kh* element was evaluated when inherited either through male or female lineages by sequentially outcrossing heterozygous Re*kh* individuals to WT mosquitoes and assessing the percentage of individuals inheriting the drive allele (i.e., transmission) and the percentage of *kh* alleles converted to Re*kh* by HDR copying (i.e., gene conversion or HDR rate).

As reported previously for the nRec element[8], near-complete drive transmission (99.8% ± 0.15% SEM, $n = 6$) and gene conversion (99.5% ± 0.29% SEM, $n = 6$) were observed in progeny whose drive-bearing parents were males (Fig. 2; Supplementary Table 2). Similar experiments with Re*kh* females showed reduced transmission (57% ± 2.2% SEM, $n = 6$) and gene conversion (14% ± 4.4% SEM, $n = 6$) (Fig. 2; Supplementary Table 2), also consistent with data reported for nRec[8]. We attribute this reduced level of drive through females to result primarily from maternal deposition of Cas9/gRNA complexes in the unfertilized egg, which accumulate to impactful levels when using the *vasa* promoter to express the Cas9 transgene[5,8,18]. A large fraction of mosquitoes that failed to inherit the drive element through the female lineage displayed a white-eye phenotype, which most likely results from early somatic mutagenesis of the WT paternal allele in eggs that inherited a

nonfunctional EJ ($kh^-$) allele from drive mothers. The combination of an inherited $kh^-$ maternal allele and a newly mutated paternal allele creates a mosaic mosquito with large sectors of homozygous or heteroallelic ($kh^-/kh^-$) mutant cells that then give rise to the eyes and other tissues.

**Re*kh* drives efficiently in caged mosquito populations.** Because loss-of-function $kh^-$ alleles are recessive, a single copy of the Re*kh* element producing a WT protein sequence under native transcriptional control is predicted to rescue *kh* activity. Consistent with this expectation, fitness assessments indicate that Re*kh* drive females are comparable to WT in their ability to reproduce (Supplementary Tables 3 and 4) and males carrying a drive allele have an equal ability to contribute to the subsequent generation as WT male counterparts (Supplementary Tables 3–5). In contrast, homozygous EJ events resulting in nonfunctional $kh^-$ alleles reduce viability, fecundity, and fertility in blood-fed females relative to WT, heterozygous, or homozygous Re*kh* females (Supplementary Tables 3 and 4). These potentially drive-resistant nonfunctional EJ alleles are expected to be eliminated from populations in cage experiments by a combination of two processes. The first mechanism, lethal/sterile mosaicism, relies on the maternally deposited Cas9/gRNA complexes acting somatically on the WT $kh^+$ paternal allele to mutate it by EJ-induced

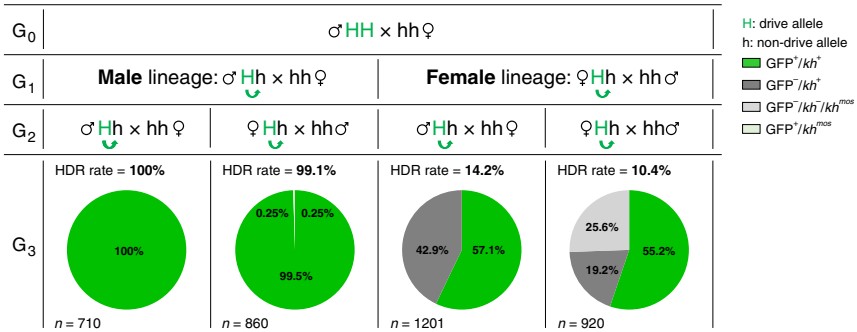

**Fig. 2 Inheritance of Reckh through paternal and maternal lineages.** Charts represent the proportion of individuals inheriting the Re*ckh* drive element (GFP⁺, in green) from heterozygous parents originating from drive males or drive females. The proportions of individuals that have not inherited the drive (GFP⁻) element and have WT black eyes (*kh*⁺) (dark grey) and those with white (*kh*⁻) or mosaic (*kh*^mos) eyes (light grey) are also shown. Rare (*n* = 2) drive individuals with mosaic eyes (GFP⁺/*kh*^mos) are depicted in light green. "H" and "h" refer to the mosquito genome at the *kh* locus, where "H" is the Re*ckh* drive allele and "h" is a non-drive allele. The green arrows show the potential for conversion of the h allele in the germline. The corresponding HDR rate, i.e., the proportion of h alleles converted to H alleles is reported. Each cross was performed en masse (30 females and 15 males) in triplicate cages using drive individuals mated to WT and by screening a representative subset of individuals (*n*) generated in the progeny. The numbers reported are pooled from the three replicate cages. Raw data for these crosses are reported in Supplementary Table 2. Data on transmission and HDR rates reported in the "Results" section refer to the averages of progeny from both mothers and fathers originating from a male or a female drive individual.

INDEL formation rendering part or all of the embryonic cells homozygous or heteroallelic for *kh* loss-of-function alleles. The second process is standard Mendelian inheritance and negative selection whereby mating of two heterozygous *kh*⁻ carriers results in 25% of the progeny carrying homozygous or heteroallelic mutant *kh*⁻/*kh*⁻ combinations. Females with these genotypes are unable to contribute effectively to the following generation. We hypothesize that the combination of fast-acting lethal/sterile mosaicism and slower-acting Mendelian processes efficiently culls *kh*⁻ alleles from the population resulting in the drive achieving full introduction (i.e., all individuals carrying at least one drive allele).

The performance of Re*ckh* in small laboratory populations was tested by assessing the drive dynamics in three sets of triplicate competition experiments (A–C) with differing drive seeding ratios. Each of these cage trials was initiated with specific release ratios of heterozygous Re*ckh* to WT males, 1:1 (initial drive allele frequency = 25%), 1:3 (initial drive allele frequency = 12.5%), or 1:9 (initial drive allele frequency = 5%) and an equal total number of WT females (1:1 sex ratio) (Supplementary Fig. 2). The proportions of eye-phenotype combinations (GFP⁺ or GFP⁻ fluorescence, and black, white, or mosaic color) were scored in a randomly selected sample of ~500 individuals for 18–20 discrete (nonoverlapping) generations (Fig. 3; Supplementary Tables 6–8).

Cages seeded with 1:1 Re*ckh*:WT males reached ≥95% introduction within 5–7 generations (Fig. 3a). Interestingly, cages 1:3_A and 1:3_C achieved ≥95% introduction within a comparable timeframe, generations 6 and 7, while 1:3_B fluctuated below 30% for 7 generations before increasing with dynamics similar to that of the other two cages and reaching >90% introduction by generation 11 (Fig. 3b). The drive dynamics in 1:3_B were evaluated further and are reported in the next section. Finally, cages seeded with the lowest release ratio (1:9) reached ≥95% introduction by generations 10 and 11 (Fig. 3c), consistent with an expected ~3-generation delay relative to cages seeded at 1:1. Estimated population levels fluctuated in all cages, but did not show any obvious decline associated with the increasing prevalence of the drive (Supplementary Fig. 3).

Potential drive-resistant individuals (GFP⁻/white) could be scored first in generation 3 mosquitoes. Sequencing their *kh* loci confirmed that the phenotype was due to both gene copies being inactivated by out-of-frame or in-frame nonfunctional heteroallelic or homozygous mutations (Supplementary Table 9). As expected from the load observed in adult *kh*⁻/*kh*⁻ females, the proportion of GFP⁻/white individuals decreased progressively over subsequent generations (Fig. 3). However, as hypothesized, the rate of decrease exceeded that expected for the elimination of homozygous recessive individuals solely by negative selection and Mendelian inheritance. We propose that this is driven by lethal/sterile mosaicism (Fig. 4a–c), which accelerates the elimination of *kh*⁻ mutations inherited through females. According to this hypothesis, perduring Cas9/gRNA complexes mutagenize WT paternal *kh*⁺ alleles somatically rendering EJ-derived loss-of-function mutations functionally dominant when transmitted by females. Because inheriting a copy of the *kh*^Rec+ allele restores WT *kh* activity, including eye color and female survival and reproductive capacity, Re*ckh* females are protected from the deleterious effects of mosaicism (Fig. 4b). In contrast, females carrying a nonfunctional EJ-derived *kh*⁻ allele often have somatic tissues comprised of double-mutant cells and therefore fail to contribute significantly to the following generation (Fig. 4c). Consistent with this interpretation, the frequency of EJ-induced *kh*⁻ alleles (scored as individuals with white eyes) was highest at the steepest phase in the drive curves reflecting the generation of mosaic individuals (Fig. 3; Supplementary Fig. 4). Evidence of somatic mutagenesis affecting small patches of tissue in the eyes was observed by the transient appearance of non-drive individuals with mosaic eyes (GFP⁻/mosaic) in generations 3–9 (Fig. 3; Supplementary Tables 6–8).

Rare (20/79,425; 0.025%) white-eyed drive (GFP⁺/white) individuals also were recovered from six cages (Supplementary Tables 6–8). Sequencing of the *kh* locus in a subset of these individuals showed that they carried drive alleles with an out-of-frame INDEL at the insertional site junction that disrupts *kh* recoding (*kh*^Rec−) in combination with a *kh*⁻ allele (Supplementary Table 10). Such *kh*^Rec− alleles may be generated by inaccurate HDR or EJ events resulting from the drive allele being targeted at low frequency by the endogenous gRNA. While such alleles may still retain drive capacity, they also suffer from the load observed in white-eyed females, consistent with these genotypes being rare and not accumulating in any of the cages. The rarity of this class of events and their failure to thrive provide clear experimental support for robust HDR-mediated rescue of endogenous gene function and validation of the recoding strategy.

When comparing the performance of Re*ckh* in population cages to previous studies conducted on the nRec drive[9]

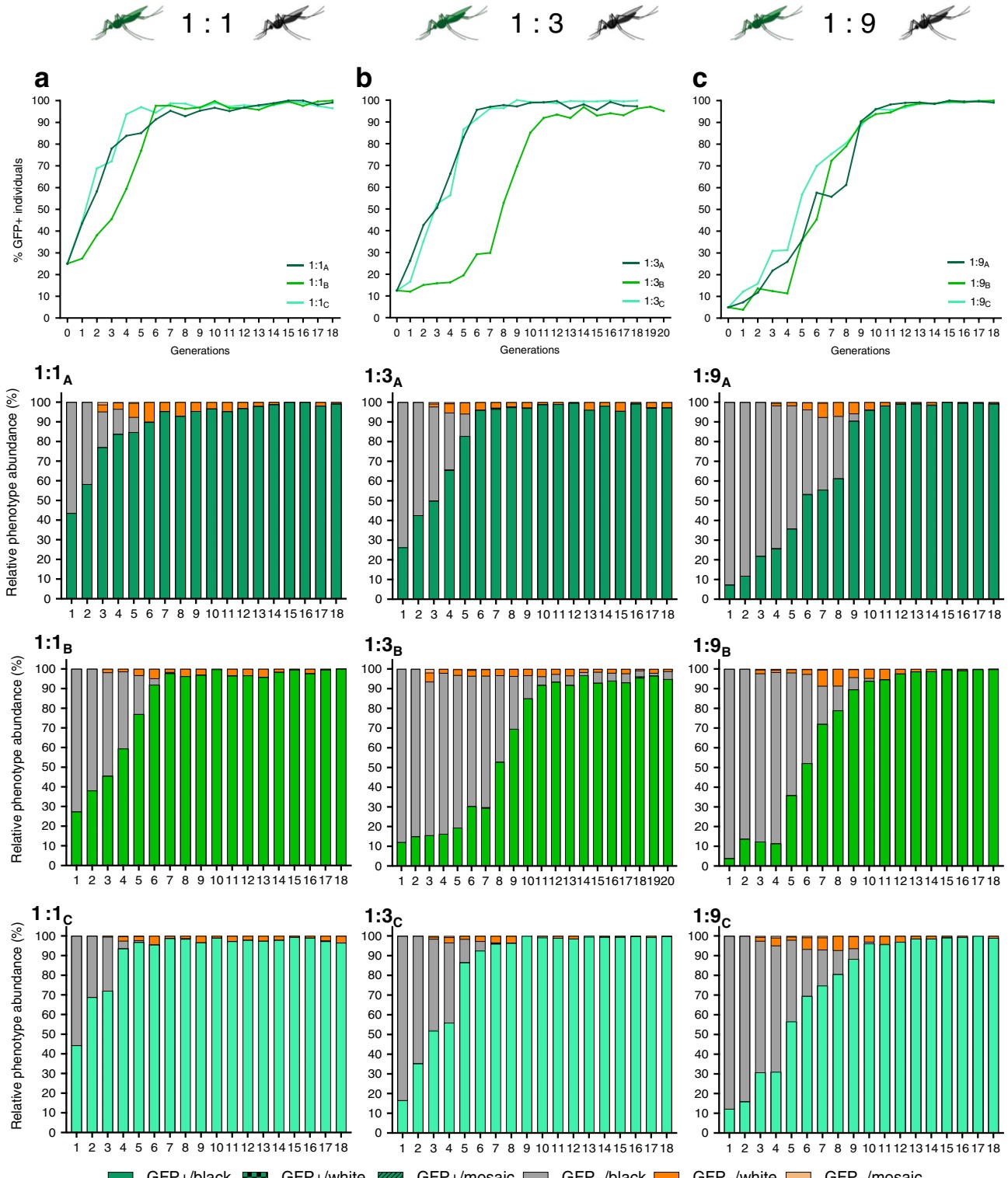

**Fig. 3 Dynamics of Reckh over 18–20 discrete generations in caged populations seeded with three release ratios of Reckh:WT males.** Top row: drive efficiency shown as percentage of GFP$^+$ individuals (Y-axes) at each generation (X-axes) in triplicate cages seeded with 1:1 (**a**), 1:3 (**b**), and 1:9 (**c**) Rec*kh*: WT male ratios. Bottom three rows: relative proportion of eye phenotypes (Y-axes) observed in a sample of ~500 individuals reported for each generation (X-axes) for all cages. Individuals containing the drive are shown in green, those with WT phenotype in grey, and non-drive individuals with white or mosaic eyes in dark and light orange, respectively. A schematic of the protocol used is reported in Supplementary Fig. 2 and raw data for each cage in Supplementary Tables 6–8.

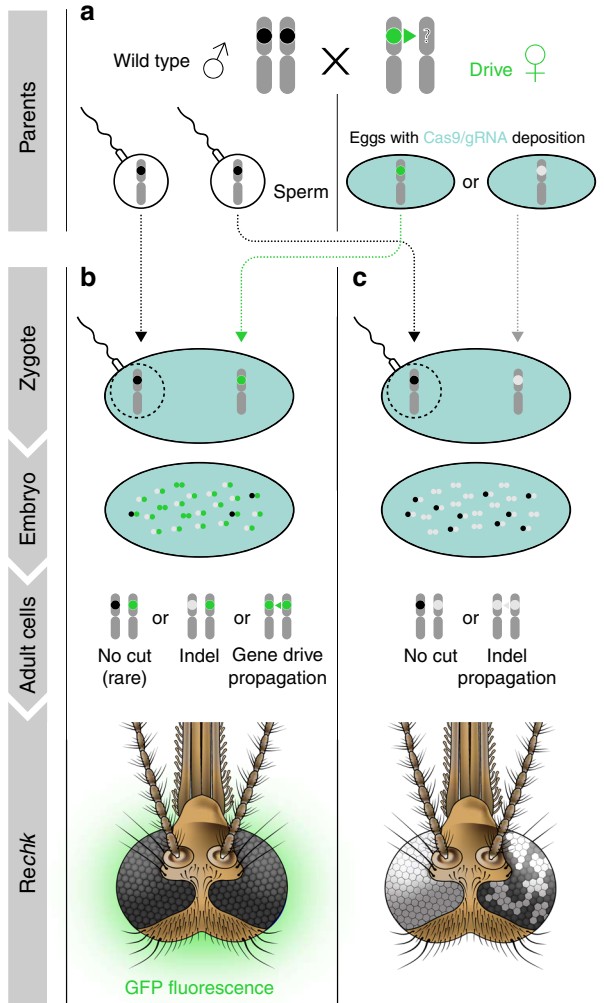

**Fig. 4 Effects of lethal/sterile mosaicism on the Re*ckh* gene-drive system. a** A female heterozygous for the drive can produce eggs carrying a copy of the drive (green circle, $kh^{Rec+}$) or eggs carrying an EJ-induced nonfunctional resistant allele (white circle, $kh^{-}$). Both types of eggs carry maternally deposited cytoplasmic Cas9/gRNA complexes (light blue filling) that can act on the incoming WT paternal allele (black circle, $kh^{+}$). **b** The soma of individuals inheriting a copy of the drive from their mothers is a mosaic of cells with varying proportions of genotypes $kh^{Rec+}/kh^{-}$, $kh^{Rec+}/kh^{+}$, and $kh^{Rec+}/kh^{Rec+}$. Re*ckh* individuals emerging from such embryos have at least one functional copy of *kh* provided by the drive system ($kh^{Rec+}$), therefore have GFP$^{+}$/black eyes and females are fit for reproduction. **c** The soma of individuals inheriting an EJ nonfunctional mutation from their drive mothers is a mosaic of cells with genotypes $kh^{-}/kh^{-}$ or $kh^{-}/kh^{+}$. The ability of females emerging from such embryos to survive and reproduce depends on the proportion of somatic cells with genotype $kh^{-}/kh^{-}$. These individuals may display mosaic or white-eye phenotype if mutations affect the cells forming the eyes. Diploid cells in (**b**) and (**c**) that become germline progenitors also may be affected by mosaicism, which can affect drive capabilities.

(Supplementary Fig. 5), we found little difference in the dynamics of the initial exponential phase of the drive curves at all seeding ratios. However, drive trajectories diverged significantly at later generations during the middle and final phases of the drive process. All Re*ckh* cages reached maximum introduction and the drive system was maintained stably for the remaining observed generations. In contrast, only two of the three 1:1 and 1:3 nRec

cages, and none of the 1:9 replicates reached >95% introduction before either the mosquito population crashed due to fixation of *kh* double-mutant genotypes or the drive was selected out of the population. A delay in mid-stage growth in the nRec 1:3 cages was observed, most likely resulting from the elimination through lethal/sterile mosaicism of female progeny receiving a $kh^{nRec-}$ or a $kh^{-}$ allele from drive mothers[9].

**A functional resistant allele does not prevent drive introduction in caged populations.** Cage replicate $1:3_B$ displayed anomalous drive dynamics where the drive stalled for the first seven generations before increasing and reaching full introduction (Fig. 3). We examined the potential basis for this outlier cage by assessing whether the presence and relative abundance of specific Cas9-induced mutations might have accounted for the observed drive delay. We deep-sequenced amplicons of the genomic DNA surrounding the cut site in non-drive alleles from pooled individuals from generations $G_0$, $G_8$, and $G_{14}$ (Supplementary Table 11).

A total of 98% of the non-drive *kh* alleles found in $G_0$ were identical to the unmutated WT sequence and the remaining were mostly INDELs adjacent to the gRNA-directed cut site. These $kh^{-}$ mutations were likely carried by the heterozygous males selected at random to seed the first cage from a founder cage that had been intercrossed for several generations. Single-nucleotide substitutions (0.03–0.2%) are consistent with the presence of rare polymorphisms around the target site in our colony cage. By the time the drive reached 50% introduction in generation $G_8$, 83% of the non-drive *kh* alleles were still unmodified WT $kh^{+}$ alleles. The most prevalent mutated non-drive alleles were INDELs of various lengths (1–3% each) and two in-frame substitutions (1–2% each) causing non-synonymous amino acid changes (TACG > CGAT: Y328R–G329W and CAG > GCA:Q330A). WT *kh* alleles were rare (<0.1%) in generation $G_{14}$ and the drive reached >95% introduction. The Y328R–G329W substitution was absent, suggesting it disrupted protein function, while the Q330A substitution had increased in frequency to 12% of the total non-drive alleles.

Further investigation of the CAG > GCA:Q330A mutation confirmed that it maintains *kh* function, as individuals homozygous for the mutation or trans-heterozygous for the mutation and a $kh^{-}$ allele isolated from generation $G_{16}$ had black eyes (Supplementary Table 12). This functional mutation affects the PAM site creating a *kh* allele that is largely (or totally) resistant to further Cas9 cleavage ($kh^{+R}$), as demonstrated by the 3:1 Mendelian segregation of the GFP marker and the absence of white-eyed or mosaic progeny in multigenerational competition cage experiments seeded with equal proportions of $kh^{Rec+}/kh^{+R}$ males and females (described in more detail below and in Fig. 5).

Resistant EJ events that preserve gene activity could be (1) under negative selection relative to the $kh^{Rec+}$ if they affect protein function, (2) positively selected if they cause a relative increase in fitness compared to drive individuals, or (3) neutral if the fitness of the two modified *kh* alleles is comparable. Population analysis of the late stages in the drive process did not support either a positive or negative selection model for the $kh^{+R}$ mutation, since the frequency of GFP$^{-}$ black-eyed individuals remained steady (1–6%) over 10 generations ($G_{11}$–$G_{20}$), consistent with a negligible fitness difference between the $kh^{+R}$ EJ allele and the recoded $kh^{Rec+}$ drive allele (Supplementary Table 7).

We further tested the relative fitness of the drive and the functional resistant alleles in a multigenerational cage experiment where $kh^{Rec+}$ and $kh^{+R}$ could compete. To do so, we monitored eye fluorescence and pigmentation phenotypes in triplicate cages

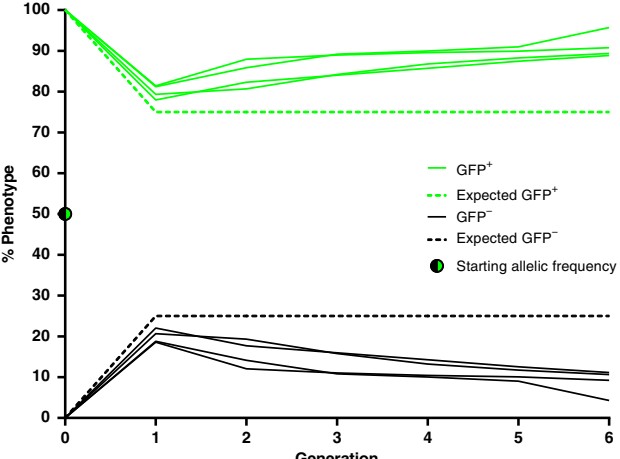

**Fig. 5 Competition between the drive $kh^{Rec+}$ and the resistant functional $kh^{+R}$ alleles in caged populations.** Four independent cages were set up using 100 male and 100 female $kh^{Rec+}/kh^{+R}$ mosquitoes each, which corresponds to an initial allelic frequency of 50%, marked using a half green-half black dot at generation $G_0$. Eye fluorescence (GFP+ or GFP−) and color (black, white, or mosaic) frequencies associated with each modified $kh$ allele were scored at every generation for six consecutive nonoverlapping generations. The proportion of GFP+ individuals (genotype $kh^{Rec+}/kh^{Rec+}$ or $kh^{Rec+}/kh^{+R}$) in the replicate cages is depicted by green lines and its expected frequency in the presence of equal competition between the two alleles (75%) as a dashed green line. The proportion of GFP− individuals (genotype $kh^{+R}/kh^{+R}$) is depicted as black lines and its expected frequency in the presence of equal competition between the two alleles (25%) as a dashed black line. All individuals screened exhibted fully WT black eyes. Raw data for these crosses are reported in Supplementary Table 13.

seeded at equal (1:1) ratios of the two alleles where $kh^{Rec+}/kh^{+R}$ individuals could mate. Assuming random mating in these conditions, phenotypes should stabilize at a 3:1 ratio of GFP+ to GFP− individuals if the two alleles are comparably competitive. The proportion of GFP−/black-eyed $kh^{+R}/kh^{+R}$ did not exceed the 25% expected for Mendelian inheritance, but in fact decreased over time (Fig. 5; Supplementary Table 13), consistent with a fitness load associated with this mutation relative to the drive allele. We found no evidence in these experiments for the resistant allele being cut at low frequency by Cas9 since we did not recover any individual with mosaic or white eyes. In light of these observations, we conclude that the appearance of the $kh^{+R}$ resistant functional allele in the 1:3_B cage is unlikely to have contributed to the pause in drive dynamics observed, and moreover that the $kh^{Rec+}$ drive allele is at least as fit, if not more so, than the functional $kh^{+R}$ EJ allele.

**Modeling Re$kh$ gene drive.** We performed mathematical modeling to assess whether the observed experimental drive dynamics in the cages conformed with predictions based on the copying efficiency of Re$kh$ in single-generation crosses and with genotype-specific loads in females. A model of autosomal Cas9/gRNA-based gene drive, similar to one used for nRec[9], was fitted to the observed cage data and includes two alternate mutated resistant alleles, functional ($kh^{+R}$) and nonfunctional ($kh^{-}$), maternal deposition of Cas9/gRNA complexes, and genotype-specific loads.

Model fitting was consistent with high (>99%) HDR efficiencies in males and females in the absence of maternal Cas9/gRNA deposition and with 17% (95% CrI: 16–18%) of the remaining EJ alleles (<1%) being functional, and the remainder being

nonfunctional. Maternal deposition was inferred to result in cleavage of embryonic WT alleles with a frequency of 93.7% (95% CrI: 92.2–95.3%), with cleavage events producing functional or nonfunctional resistant alleles having the same distribution to preserve identifiability in the model-fitting process. Data also are consistent with $kh^{-}/kh^{-}$ females having a reproductive load of 99.8% (95% CrI: 99.2–100%), while the drive allele is associated with a negligible load, consistent with laboratory observations. The trajectories of GFP+ individuals align well with experimental observations at all release ratios and are consistent with a highly efficient gene-drive system (Fig. 6, Supplementary Fig. 6).

Functional resistant alleles are generated at a low rate, and although they can persist in the population due to their resistance to cleavage (GFP−/black in Supplementary Fig. 6), they were not a significant obstacle to the drive at the release ratios analyzed. Nonfunctional resistant alleles also were generated at a low rate and were strongly selected against in the progeny of females that generated them, while their subsequent elimination was gradual (GFP−/white in Supplementary Fig. 6) due to their viability in males and heterozygotes of both sexes.

Finally, a stochastic model captured the potential role of chance events such as mate choice (multinomial distributed), egg production (Poisson distributed), progeny genotype (multinomial distributed), and finite sampling of the next generation (multivariate hypergeometric distributed) (Fig. 6). Stochastic model trajectories reflect some of the variability observed in the early stages of spread of the gene-drive allele, with additional transient delays observed in cages 1:1_B and 1:3_B.

## Discussion

Population modification strategies employing gene-drive systems to spread anti-parasite effector molecules through populations of *Anopheles* mosquitoes are gaining momentum in the fight against malaria[10,30,31]. However, the creation of mutated target sequences resistant to the drive, especially those preserving gene function, can limit transgene introduction, particularly if a load is associated with the presence of the drive system[9,13].

Here, we provide experimental evidence for a readily generalizable gene-drive rescue system for efficient population modification in *Anopheles* mosquitoes that actively eliminates nonfunctional resistant alleles as they arise and rapidly attains >95% introduction in caged populations, despite the creation of rare functional EJ variants. As a result, the Re$kh$ system converts a population suppression gene-drive (nRec) to an efficient population modification system in *An. stephensi*. Furthermore, while proof-of-concept for CRISPR/Cas9-based rescue systems that employ recoded sequences has been produced in *D. melanogaster*[20,21,24], this work reports the application of such a system to mosquitoes. Assessments of the long-term dynamics of this system in caged mosquito populations show that Re$kh$ spreads quickly (5–11 generations depending on release ratio) and efficiently even when a small percentage of males are released. After reaching maximum introduction, the Re$kh$ drive persists without an evident impact on the population size.

Re$kh$ targets the haplosufficient gene $kh$, required for adult female survival and reproduction in *An. stephensi* following a blood meal[9], and provides a recoded portion of the gene that rescues its function. In doing so, individuals that carry a copy of the drive are functionally protected and comparably as fit as their WT counterparts, while nonfunctional resistant alleles are eliminated owing to the reduced survival and impaired reproductive capacity in white-eyed homozygous and mosaic heterozygous females. We propose that such elimination is driven initially by the active process of lethal/sterile mosaicism[25], during which recessive nonfunctional $kh^{-}$ mutations function as

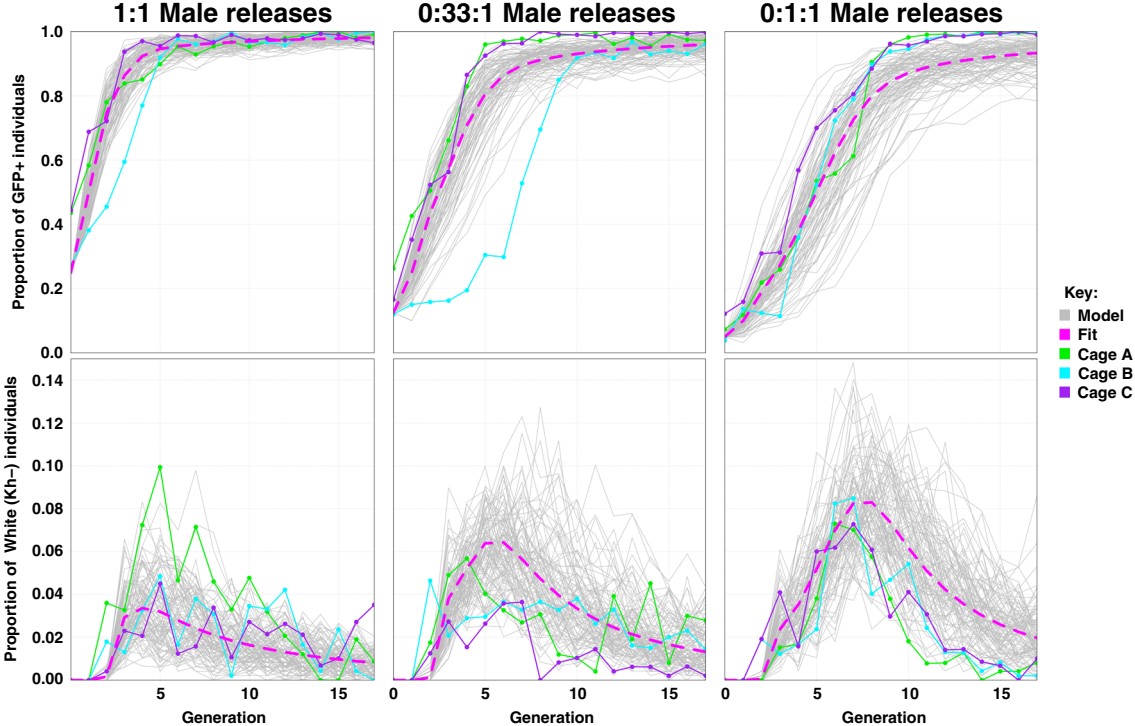

**Fig. 6 Observed and model-predicted dynamics of GFP$^+$ and $kh^-$ phenotypes in the Reckh cage experiments.** Solid green, blue, and purple lines represent the experimental data over 18 generations observed in 3 replicates (Cages A–C) with release ratios of Rec*kh*:WT males of 1:1, 1:3, and 1:9, respectively. Dotted pink lines represent the fitted deterministic model (Fit), and grey lines are 100 stochastic realizations of the fitted model for each release ratio (Model). X-axes report the generation number after release and Y-axes the proportion of each eye phenotype. The GFP$^+$ phenotype results from having at least one copy of the drive allele and hence reflects the spread of the gene-drive system to full or near full introduction for all experiments. The $kh^-$ phenotype is associated with having no copies of the WT, Rec*kh*, or functional resistant alleles (i.e., having two copies of the nonfunctional resistant allele) and reflects the low-level emergence and gradual elimination of this allele from the population due to its load in homozygous females. The stochastic model captures the variability inherent in the experimental process and reflects some of the variability observed in the early stages of the spread of the gene-drive allele.

dominant alleles during the drive process, therefore acting as an autocatalytic mechanism that eliminates female progeny of drive mothers from the breeding pool. Mosaicism-mediated elimination is complemented by negative selection of the residual $kh^-$ recessive alleles transmitted through male progeny, which acts over many generations to eliminate these costly alleles from the population. Overall, the accelerated elimination of $kh^-$ alleles results in an increased apparent HDR-mediated conversion frequency.

Modeling predictions based on transmission frequencies and loads of different allele combinations are consistent with the highly efficient gene-drive outcomes observed in the cage experiments. Simulations are consistent with nonfunctional $kh^-$ resistant alleles, which are generated primarily in females, being eliminated rapidly by lethal/sterile mosaicism and then gradually as standard recessive alleles due to the load associated with bi-allelic $kh$ loss-of-function. Consistent with lethal/mosaicism taking place in offspring of gene-drive females, the presence of white-eyed individuals carrying $kh$ homozygous mutations in the cage experiments followed a general trend in which they were most abundant at the steepest phase in the drive curve, while the remaining $kh^-$ alleles propagating through males were slowly eliminated through negative selection over time. Potential loads in white-eyed males, such as impaired vision, also could have contributed to the elimination of nonfunctional resistant alleles. Future recoded drive systems inserted into genes essential for viability of both sexes or required for both male and female fertility could drive even more rapidly.

The lag in drive invasion observed in one of the 1:3 release ratio cages, which deviates from the consistent trends observed in the other eight cages, remains difficult to explain since it is not fully supported by the predictions of the stochastic model. It is not likely to derive from a sampling effect, as a delay in the drive increase was not observed in any other cage, including those seeded with the lowest release ratio where such effect, if present, would be more prominent. Also, our analysis does not support involvement of resistant functional alleles in this observation. Further analysis of these trajectories will be required to ascertain the underlying mechanisms responsible for such outlier events. Nevertheless, the transient delay in drive invasion observed in this cage did not prevent eventual maximum introduction of the drive.

A major concern associated with the persistence of gene-drives is the generation of cleavage-resistant sequences that preserve gene function. In population suppression systems, these alleles are positively selected over the drive allele if the latter causes loss-of-function of the targeted gene resulting in a fitness disadvantage[9,13]. Therefore, choosing a functionally constrained target site should alleviate the issue of resistance and this proved successful in suppression and sex-distorting strategies[6,7]. This approach also can mitigate the effect of resistance in population modification systems, where the effect of functional resistant target sequences on the drive dynamics is dependent on their fitness relative to that of the drive. Data from the cage experiments and modeling show that functional resistant alleles in the Rec*kh* system are generated at a low rate and do not prevent the

drive from reaching full introduction. We attribute this outcome to the high functional constraint of the chosen target site within the *kh* gene that is likely to cause total or partial protein malfunction when mutated. This hypothesis is supported by evidence from the crystal structure of the KH enzyme from *Saccharomyces cerevisiae* where the P326–F327–Y328–G329–Q330 loop, which encompasses the gRNA site in our experiments, plays an important role in accommodating the rearrangements of the active site of the enzyme during binding[32]. Indeed, this region shows high primary amino acid sequence conservation from insects to humans[28,33]. Furthermore, we found examples of in-frame mutations in individuals with white eyes, consistent with previous findings from similar cage experiments conducted using the same gRNA where the strong contribution of the Y328 and G329 residues to protein function was highlighted[9]. Here we report evidence for the contribution of the Q330 residue, whose codon forms part of the PAM site. While individuals homozygous for the Q330A substitution display a black WT eye phenotype and the mutation is resistant to Cas9 cleavage, we observed only a modest accumulation of this mutation amongst the non-drive alleles that did not prevent drive invasion. We explored this further in cage experiments were the mutated resistant allele was in direct competition with the drive and found that the mutation was outcompeted by the drive. We therefore conclude that the Q330A mutation likely only partially restores WT levels of KH enzyme and thus does not pose a major obstacle to the drive process. Given the documented functional constraints of multiple amino acid residues at the *kh* target site and evidence that the $kh^{Rec+}$ allele does not carry an obvious fitness burden, it is likely that this drive will outcompete a number of Cas9-induced functional resistant mutations. Similarly, while we currently do not have access to genome data on field populations of *An. stephensi*, we would not expect the targeted conserved coding sequence to show high levels of natural polymorphism. Nevertheless, analysis of the *kh* locus in field-caught *An. stephensi* natural populations is needed to assess this question.

Field performance of population modification drives is not only dependent on the drive copying process as such but also on effects on mosquito fitness produced by sequences encoded in the cargo (e.g., the antimalarial effectors). The Re*kh* construct currently lacks antimalarial molecules as the two effectors m2A10 and m1C3 present in the original prototype were excised due to molecular constraints to allow for partial *kh* recoding and simultaneous fluorescent marker exchange. However, we do not anticipate major fitness impacts in our drive mosquitoes as our population modification strategy is based on the use of synthetic single-chain antibodies (scFv) specifically directed against *Plasmodium* parasite antigens. Due to their target specificity and regulated blood meal-induced expression, combinations of these molecules have been shown to have little (if any) impact on mosquito fitness[11]. In contrast, expression of multiple toxins and synthetic molecules with broader activity can exert undesired impacts on crucial physiological processes or the gut microbiota[34]. Nonetheless, comprehensive fitness assessments in insectary cages and contained field trials are needed to validate these hypotheses. Furthermore, we predict that the addition of genes coding for antimalarial effectors and their regulatory regions will not greatly affect drive copying since the initial exponential growth of Re*kh* and nRec systems were similar, supporting the conclusion that Re*kh* will tolerate an increase in cargo size (i.e., the nRec construct is ~4.6 kb larger than Re*kh*).

We also report the application to mosquitoes of the highly efficient Swap transgenesis technology, a Cas9/gRNA-driven cassette exchange system[29], to update sequences of previously integrated gene-drive systems. Swap is a flexible tool that efficiently edits existing gene-drive mosquito lines using small

constructs. Swap also can enable broader genome engineering efforts in mosquitoes. Since it does not require the presence of recombination sites such as those needed for *φC31* recombinase-mediated cassette exchange, updating sequences could be inserted anywhere suitable gRNA sites are available. For example, the system could be used to exploit endogenous *cis*-acting elements[35,36] through the seamless integration of desired coding regions. For population modification purposes, a drive line can be envisioned that carries strategically placed gRNA targets to replace antiparasitic molecule combinations and test their blocking efficacy. Indeed, our next steps include the addition of combinations of antimalarial scFvs that block parasite development using Swap. While the Re*kh* line can be used as a classic docking line via the *φC31 attP* site introduced along with the recoded *kh* sequence, integration via this site would significantly increase the size of the sequence between the homology arms as well as bring in additional sequences that might impact drive performance. Nevertheless, this efficient integration strategy can be used to assess blocking capabilities of alternative combinations of antimalarial effectors expressed at the *kh* genomic locus. Finally, while the Re*kh* system described here is already highly efficient, substituting the current *vasa* promoter with more tightly regulated control sequences, such as those of the *nanos*[10] or *zpg*[18] genes employed for gene-drive designs in *An. gambiae*, could improve its performance by reducing the fraction of EJ alleles persisting during the drive process. However, we believe that lethal-mosaicism provides a general solution to the problem of females generating EJ alleles and should allow for the design of efficient gene drives at many different target loci and insect species where identification of "ideal" promoters may not be straightforward or transferable.

Overall, the laboratory assessments conducted so far, carried out in line with the recommended phased pathway for testing gene-drive mosquitoes[37–39], show that the characteristics of the Re*kh* gene-drive are likely to conform with those defined as part of a proposed Target Product Profile for population modification of mosquito strains[30] once the addition of the anti-malaria effectors is proven effective under this configuration. The highly efficient performance of the Re*kh* drive system makes it an excellent candidate for genetic control of an important malaria mosquito, *An. stephensi*, and this technology should be readily adaptable to other mosquito species as well as other insect disease vectors.

## Methods

**Mosquitoes**. *An. stephensi* Indian strain (gift from M. Jacobs-Lorena, Johns Hopkins University) were maintained in insectary conditions (27 °C and 77% humidity) with a photoperiod of 12-h light/dark and 30 min of dawn/dusk. Sucrose solutions (10% wt/vol) were provided ad libitum and blood meals consisting of defibrinated calf blood (Colorado Serum Co., Denver) were offered to 3–7-day-old adults through the Hemotek® membrane feeding system. Larval stages were reared in distilled water and fed TetraMin® fish food mixed with yeast powder. Gene-drive mosquitoes were contained in ACL-2 insectary facilities at the University of California, Irvine and handled according to recommended safety procedures[40–42].

**Plasmids for the Swap strategy**. The Swap strategy employed to convert nRec to Re*kh* uses the three plasmids shown in Fig. 1a: (1) pVG362_Aste-U6A-Swap3-gRNA to express the gRNA-sw3, (2) pVG363_Aste-U6A-Swap4-gRNA to express the gRNA-sw4, and (3) pVG344_Aste_kh2-MCRv3-vasa-Cas9 to provide the HDR template containing the recoded-*kh* coding fragment and the GFP marker.

To generate plasmids pVG362 and pVG363, a pair of oligonucleotides were synthesized (Integrated DNA Technologies) for each plasmid with 19 (pVG362) or 20 (pVG363) bases of the target sequence chosen for the strategy. These were annealed and ligated with T4 ligase (New England Biolabs) into the pVG145-Aste-U6A-Bbs1 plasmid[8] linearized with *Bbs*I. The cloning strategy was adapted from the work of Port et al.[43]. The oligonucleotides used to construct pVG362 were 1288_Aste-Swap3-Target-F (CTTGTTCTTGGAGGAGCGCACCA) and 1289_Aste-Swap3-Target-R (AAACTGGTGCGCTCCTCCAAGAA). The oligonucleotides used to construct pVG363 were 1290_Aste-Swap4-Target-F

(CTTGTTACGttaattaaCGTAGAA) and 1291_Aste-Swap4-Target_R (AAACTTCTACGttaattaaCGTAA).

The pVG344 plasmid was cloned using the NEBuilder HIFI DNA Assembly Cloning Kit (New England Biolabs) to assemble four amplified fragments. Fragment 1 was generated by amplification of the backbone region of plasmid pVG163_pAsMCRkh2[8], fragment 2 was generated by amplification of the *kh* recoded rescue fragment from a plasmid synthesized by GenScript Inc., fragment 3 was amplified from a plasmid containing a 3× P3-GFP cassette commonly used for insect transgenesis, and fragment 4 also was amplified using pVG163_pAsMCRkh2 as a template. Primer pairs used to amplify each fragment were: Fragment 1: 494_pUC19_Backbone_F (GGTATCAGCTCACTCAAAGGCGGTAATACGG) and 1227_As-MCR2_GA_backbone_R (CGTAGAACGGAACCATCGCGTG), Fragment 2: 1231_As-MCR2_GA_RecodedFrag_F (CGCGATGGTTCCG TTCTACGG) and 1232_As-MCR2_GA_RecodedFrag_R (CTACGCCCC,, CAACTGAGAGAACTC), Fragment 3: 1230_As-MCR2_GA_GFP_F (TCTCTCAGTTGGGGGCGTAGCGTACGCGTATCGATAAGCTTTAAGA-TAC) and 1229_As-MCR2_GA_GFP_R (CACCGGTCGCCACCATGGTGA GCAAGGGCGAGGAGCTGTTCAC, Fragment 4: 1228_As-MCR_GA_backb one_F (CACCATGGTGGCGACCGGTGGATC) and 1241_As_MCR2_GA_ HA2_R (CGCCTTTGAGTGAGCTGATACCGTGAGCAAAAGGAGACGG).

**Microinjections and establishment of Re*ckh*.** Embryos were obtained from heterozygous females of the *An. stephensi* AsMCRkh2 (nRec) gene-drive line[8]. Microinjection procedures[44] were performed using a plasmid mix containing 600 ng/μL of pRec-*kh* donor (pVG344) and 200 ng/μL of each gRNA-sw3 (pVG362) and gRNA-sw4 (pVG363) plasmids. Surviving G$_0$ adults were sorted in pools of 2-4 males and 7-10 females and outcrossed to 10x WT females and 1× WT males, respectively. G$_1$ progeny were screened as larvae for the inheritance of the GFP eye marker and kept as separate lines according to their male (♂4) or female (♀4) founder lineage. The two lines were screened routinely as larvae for the inheritance of the GFP eye marker and as pupae for the eye-color phenotype (black [WT], white, or mosaic) and maintained by intercrossing GFP$^+$ black-eyed individuals. A homozygous drive line was established from the ♀4 intercrossed line.

Molecular confirmation of HDR-mediated target site integration of the Re*ckh* cargo was performed on genomic DNA extracted from single GFP$^+$ black-eyed individuals using the Wizard® genomic DNA purification kit (Promega). Primers Kh1-ext-fw (CACTGTTGGCACTCCATCTG) and Rec-kh-rv2 (GGGCTTCAAC AACTGAAAAG) were used to amplify a 2190 bp region spanning the cut site of gRNA-sw4, while primers eGFP-fw (AAGTCGTGCTGCTTCATGTG) and Vasa-rv (GTAAAAGCCGCATTTTCCAA) were used to amplify a 2303 bp region across the cut site of gRNA-sw3. Gene amplification reactions were performed using Phusion® High-Fidelity PCR Master Mix (New England Biolabs). Sanger sequencing (Genewiz, San Diego) with primers Rec-kh-rv2 and eGFP-fw was used to confirm the sequence of the integration sites.

**Primary drive transmission assessments.** Drive transmission and HDR conversion rates through the male and female lineages were assessed in sequential en masse outcrosses of Re*ckh* individuals to WT. Each cross comprised 30 females and 15 males and was performed in three replicate cages. A representative subset of the progeny of each cross was scored for the presence of the GFP fluorescent marker and the eye color phenotype (black, white, or mosaic) in adults. A schematic of the crossing performed is reported in Fig. 2.

Drive transmission is defined as the percentage of individuals inheriting the Re*ckh* element. Gene conversion or HDR rate is defined as the percentage of *kh* alleles converted to Re*ckh* by HDR copying and is calculated using the formula $[2(X - 0.5n)/n]$ ("X" is the number of GFP$^+$ individuals and "n" the total number of mosquito counted)[8].

**Female fecundity and fertility.** Homozygous Re*ckh* ($kh^{Rec+}/kh^{Rec+}$), heterozygous Re*ckh* carrying a copy of the drive and a $kh^-$ allele ($kh^{Rec+}/kh^-$), WT ($kh^+/kh^+$), and white-eye ($kh^-/kh^-$)[9] females were included in this analysis. Adult females 5–7 day old were offered a blood meal for 45 min over 2 consecutive days and unfed females removed. After 3 days, single females were set up to oviposit in 16 oz (~454 cm$^3$) paper cups containing a plastic oviposition cup lined with damp filter paper. Eggs were counted the next day using a stereomicroscope and transferred to water cups lined with filter paper for hatching. Larvae emerging from single egg batches were counted at the first or second instar (L$_1$–L$_2$). Fecundity refers to the number of eggs laid by a single female and fertility to the proportion of larvae hatching from these individual egg batches. A one-way ANOVA with Tukey's multiple comparison post hoc test was used to assess significant differences ($p > 0.05$) in the performance of females from the four groups tested.

**Male contribution to the following generation.** Triplicate cages were seeded with 75 Re*ckh* homozygous males, 75 WT males, and 150 WT females. All individuals were added to the cage as 3–7-day-old adults and females were offered a blood meal over two consecutive days. Approximately, 2000–2500 L$_4$ larvae were selected randomly from the progeny of each replicate cage and scored for the presence of

GFP. A two-tail binomial test was used to compare the observed and expected distributions and test for significant ($p > 0.05$) deviation from equal frequency (50%) of GFP$^+$ and GFP$^-$ individuals.

**Cage trial set up and maintenance.** A schematic representation of the cage trial protocol implemented is shown in Supplementary Fig. 2. The trial consisted of 18 nonoverlapping generations and was conducted in 5000 cm$^3$ cages essentially as described by Pham et al.[9]. Triplicate cages (A–C) were seeded with three single release ratios, 1:1, 1:3, 1:9, of 3–5 day old age-matched Re*ckh* heterozygous to WT (Re*ckh*:WT) male adults (100 in total), and 100 WT adult females were added to reach an equal sex ratio for a total of 200 individuals per cage. The number of Re*ckh* males was 50 in the 1:1 release cages, 25 in the 1:3 cages, and 10 in the 1:9 cages. Adults 5–7 days old were offered a blood meal over two consecutive days. Three days later, dead adults were removed from each cage and an egg cup was provided for two days. Of the hatching larvae: 200 L$_1$–L$_2$ were selected at random and reared to adulthood to establish the following generation, 500 L$_1$–L$_2$ were selected randomly to assess the progression of the drive by screening the eye phenotype in L$_4$ larvae and pupae (see "Screening of the eye phenotype" section), and the remainder reared to L$_4$ and stored in ethanol for population counts and molecular analysis. Due to the small initial number of transgenics, the only exception to the random selection of the 200 L$_1$s to seed the following generation was that individuals from generation 1 of the 1:9 cages were all screened for their eye phenotype and new cages were seeded with the same proportion of drive individuals found. Finally, the screening of cage 1:3$_B$ was carried out for additional two generations for a total of 20 generations.

**Screening of the eye phenotype.** A sample of 500 randomly selected L$_4$ larvae were scored at each generation for the presence of GFP fluorescence (GFP$^+$ and GFP$^-$) and separated in two corresponding larval trays; pupae emerging from each tray were screened for eye color (black, white, or mosaic). The phenotypes were reported as follows: GFP$^+$/*kh*$^+$ (drive individuals with black eyes); GFP$^+$/*kh*$^-$ (drive individuals with white eyes); GFP$^+$/*kh*$^{mos}$ (drive individuals with mosaic eyes); GFP$^-$*kh*$^+$ (non-drive individuals with black eyes); GFP$^-$/*kh*$^-$ (non-drive individuals with white eyes); GFP$^-$/*kh*$^{mos}$ (non-drive individuals with mosaic eyes). Among these, individuals with phenotypes GFP$^+$/*kh*$^-$ and GFP$^-$/*kh*$^-$ were stored as adults at −20 °C for sequencing.

**Population count.** L$_4$ larvae collected throughout the experiments were stored in 50 mL conical centrifuge tubes filled with ethanol. Before counting, ethanol was rinsed off and larvae were re-suspended in a fixed volume of deionized water and placed onto a shaker moving at a constant speed. Larvae were collected using a fixed-volume scoop and counted before returning them to the shaker. A total of 6–9 measurements were taken per cage every two generations. The estimated population size was calculated by averaging the number of larvae from replicate measurements and multiplying by the conversion factor (volume of water/scoop volume). The only exception to this method of counting was generation 1 of the 1:9 cages where the whole L$_4$ population was counted.

**Sanger sequencing on single mosquitoes.** Genomic DNA was extracted from whole single adult mosquitoes using either the Wizard Genomic DNA Purification Kit (Promega) or the DNeasy Blood & Tissue Kit (Qiagen). All gene amplification reactions were performed using the Phusion High Fidelity PCR Master Mix with HF Buffer (New England Biolabs). To analyze the non-drive allele, primers KhE5-4 (GACGGTGACACTGTTCATGC) and KhE5-3 (CAGATGGCATGTGCATCC TC) were used to generate a 372 bp amplicon spanning the gRNA-directed cut site in the *kh* gene. Sanger sequencing (Genewiz, San Diego) of the non-drive amplicons was performed using primer KhE5-4. To analyze the Re*ckh* drive allele, primers KhE4 (CGTTCGAGTAGCACGTTG) and Agam3 rv (CAGGTGTAGAA GAAAACACGTTG) were used to produce a 1287 bp amplicon. Sanger sequencing (Genewiz, San Diego) of the Re*ckh* amplicon was performed using primer KhE4. Sequencing results from mixed traces were resolved using CRISP-ID (http://crispid. gbiomed.kuleuven.be)[45].

**DNA extraction and amplification from pooled mosquitoes.** Genomic DNA from individuals used to seed the 1:3$_B$ cage (generation 0) was extracted from pools of 20 adults (total of ~140); while DNA from individuals from the same cage at generations 8 and 14 was extracted from pools of 50 larvae (total of 300 each). Extractions were performed using the DNeasy Blood & Tissue Kit (Qiagen) according to manufacturer's protocol with an overnight initial lysis step. An equal volume of genomic DNA was pooled from each replicate extraction and used as template for amplification. Gene amplification was performed using the Phusion High Fidelity PCR Master Mix with HF Buffer (New England Biolabs) and primers KhE5-4 (GACGGTGACACTGTTCATGC) and KhE5-3 (CAGATGG-CATGTGCATCCTC). Generated amplicons were purified from 1% agarose gels using the Zymoclean Gel DNA Recovery Kit (Zymo Research) before library preparation.

**Library preparation and sequencing.** Illumina libraries were prepared for each of three samples ($G_0$, $G_8$, and $G_{16}$ from cage $1:3_B$) using the NEXTFLEX PCR-free library preparation kit and NEXTFLEX Unique Dual Index Barcodes (BIOO Scientific) following the manufacturer's instructions. The input amount of DNA was 500 ng. The ends of the DNA were repaired and adenylated. The reaction was cleaned using AMPure XP magnetic beads and Illumina barcoded adapters were ligated onto the blunt-end adenylated product. The adapter-ligated product was cleaned using AMPure XP beads. DNA quantity was measured by Qubit DNA HS assay and the fragment size assessed by Agilent Bioanalyzer 2100 DNA HS chip assay at the genomics facility of the University of Utah (GNomEx) where the libraries were sequenced on the Illumina NovaSeq with the SP flowcell $2 \times 250$ paired end.

**Sequencing data analysis.** The raw paired-end Illumina reads from the amplified genomic region were cleaned for low quality and trimmed for the presence of adapters using Trimmomatic v0.35[46]. High-quality reads were mapped against the amplicon sequence using BWA-MEM v0.7.8[47] and the alignments sorted using SAMtools v1.9[48]. Mapped paired-end reads were extracted using Picard Tools v1.96 (http://broadinstitute.github.io/picard/), and then joined to reconstruct the complete amplicon sequence using PEAR v0.9.8[49]. Identical amplicon sequences were clustered using module fastx_collapser in FASTX-ToolKit v0.0.14 (http://hannonlab.cshl.edu/fastx_toolkit/). Clustered sequences were aligned to the reference amplicon with MAFFT v7[50] under FFT-NS-2 tree-based progressive method with 1PAM/K = 2 scoring matrix, if they were represented by ≥3 paired-end reads in each dataset. After alignment with the amplicon, the analysis was focused on a 34 bp target sequence including the 23 bp of the gRNA to identify single-nucleotide polymorphisms and/or INDELs in this region. The final quantification of mutations at the target site was measured as relative frequency of paired reads in sequence variants represented in at least 100 reads.

**Functional resistant allele assessments.** Individuals carrying a mutated functional $kh$ allele ($kh^{+R}$) due to the presence of a CAG > GCA-Q330A substitution affecting the PAM site were isolated from non-drive black-eyed (GFP⁻/black) individuals from cage $1:3_B$ at generation $G_{16}$. Resistance of the $kh^{+R}$ allele to Cas9-induced cleavage was assessed in the progeny of the cross between males heterozygous for a copy of the $Reckh$ drive allele and the $kh^{+R}$ allele ($kh^{Rec+}/kh^{+R}$) to WT females by scoring the frequencies of GFP⁺ and GFP⁻ mosquitoes (expected to be ~50% in case of resistance to cleavage).

Allele competition experiments were conducted in four replicate cages (A–D) each seeded with 200 individuals heterozygous for a copy of the $Reckh$ drive allele and a copy of the $kh$ functional resistant allele ($kh^{Rec+}/kh^{+R}$) with a 1:1 sex ratio. Allele competition was inferred from the eye phenotype of the progeny of these crosses by scoring for the presence of the GFP fluorescent marker (GFP⁺ or GFP⁻) and the eye color (black, white, or mosaics) in adults. This was carried out for six discrete (nonoverlapping) generations by screening a representative sample of ~300 adults at each generation and seeding new cages with 200 randomly picked individuals, as described for the gene-drive cage trials. In this set-up, assuming random mating, equally competitive modified $kh$ alleles are expected to maintain a phenotypic ratio of 3:1 GFP⁺:GFP⁻ individuals in each generation; deviations from this ratio would signify unequal competitiveness.

**Modeling of cage population dynamics.** Empirical data from the nonoverlapping gene-drive experiments were used to parameterize a model of Cas9/gRNA-based homing gene-drive including resistant allele formation, and a stochastic implementation of the fitted model was used to qualitatively compare the time series of observed genotype frequencies to model-predicted ones. Model fitting was carried out for all nine gene-drive cage experiments using Markov chain Monte Carlo methods in which estimated parameters related to loads, resistant allele generation, and the consequences of maternal deposition of Cas9/gRNA complexes were used.

We considered discrete generations, random mixing, and Mendelian inheritance rules at the gene-drive locus, with the exception that for adults heterozygous for the homing allele (denoted by "H") and WT allele (denoted by "W"), a proportion, $c$, of the W alleles are cleaved, while a proportion, $1 - c$, remain as W alleles. Of those that are cleaved, a proportion, $p_{HDR}$, are subject to accurate HDR and become H alleles, while a proportion, $(1 - p_{HDR})$, become resistant alleles. Of those that become resistant alleles, a proportion, $p_{RES}$, become in-frame, functional, cost-free resistant alleles (denoted by "R"), while the remainder, $(1 - p_{RES})$, become out-of-frame, nonfunctional, or otherwise costly resistant alleles (denoted by "B"). The value of $p_{HDR}$ is allowed to vary depending on whether the HW individual is female or male, and values for female- and male-specific HDR parameters were estimated based on $G_0$ crosses that provided direct information on them.

The effects of maternal deposition of Cas9/gRNA complexes were accommodated after computing the gene-drive-modified Mendelian inheritance rules. If offspring having a W allele had a mother having the H allele, then this would lead to Cas9/gRNA complexes being deposited in the embryo by the mother,

possibly resulting in cleavage of the W allele. We considered cleavage to occur in a proportion, $p_{MC}$, of these embryos, with a proportion, $p_{MR}$, of the cleaved W alleles becoming R alleles, and the remainder, $(1 - p_{MR})$, becoming B alleles.

These considerations allow us to calculate expected genotype frequencies in the next generation, and to explore the impacts of loads and maternal deposition parameters that maximize the likelihood of the experimental data. Estimated parameters include loads in females associated with having one or two copies of the H allele or the BB genotype, and $p_{RES}$, $p_{MC}$, and $p_{MR}$, as defined earlier. A stochastic version of the fitted model was implemented using a discrete generation version of the Mosquito Gene-drive Explorer (MGDrivE) model[51] with an adult population size of 200. The complete modeling framework is described in the "S1 Text" section of Pham et al.[9].

**Statistical analysis.** Statistical tests were performed as detailed in the relevant method sections using GraphPad Prism 8.4.2.

**Reporting summary.** Further information on research design is available in the Nature Research Reporting Summary linked to this article.

## Data availability
The full sequences of the plasmids used in this work are deposited in GenBank under accession numbers MW030449 for pVG362_Aste-U6a-Swap3-gRNA, MW030450 for pVG363_Aste-U6a-Swap4-gRNA, and MW030448 for pVG344_Aste_kh2-MCRv3-Vasa-Cas9_SWAP. Raw sequencing data is available in the Sequence Read Archive (SRA) database under BioProject PRJNA607757 and accession numbers SAMN14145944 for cage $1:3_B$, generation $G_0$, SAMN14145945 for cage $1:3_B$, generation $G_8$, and SAMN14145946 for cage $1:3_B$, generation $G_{14}$. A Source Data file is provided for raw data displayed in Supplementary Figs. 1, 3, and 4 and Supplementary Table 3. All other raw data are contained within the paper main text and Supplementary information. Any other relevant data is available from the authors upon reasonable request. Source data are provided with this paper.

## Code availability
The complete modeling framework is described in the S1 Text section of Pham et al.[9]. Code can be accessed by contacting J.M.M. (john.marshall@berkeley.edu).

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

## Acknowledgements
We are grateful to the members of the UCI insectary staff, Drusilla Stillinger, Kiona Parker, Sean Firth, Laura Partida, and Jazmin Murphy, as well as Savannah Bogus (UCSD), who provided excellent support in mosquito screening and husbandry. We thank Kiona Parker (UCI) for providing microscopy images of transgenic mosquitoes, Vinaya Shetty (UCI and TIGS-India) for assistance with life-table experiments, and Thai Binh Pham (UCI) for the gift of primers KhE5-4 and KhE5-3. We are also thankful to Peter Shum (Stanford University) for technical assistance with amplicon sequencing. This work was supported in part by the Tata Institute for Genetics and Society (TIGS) and the National Institutes of Health (NIH) (AI29746). V.M.G. was supported by a grant from the Office of the Director of the National Institutes of Health (DP5OD023098). J.M.M. was supported by funds from the UC Irvine Malaria Initiative. J.B.B. and J.M.M. were supported by a DARPA Safe Genes Program Grant (HR0011-17-2-0047). J.J.E. was supported by the NIH (R01GM123303). A.A.J. is a Donald Bren Professor at the University of California, Irvine.

## Author contributions
V.M.G., E.B., A.A.J., and A.A. conceived the experimental approach; A.A., N.J., H-F.L., K.H., G. T., and E.A.B performed the laboratory research; A.R. and J.J.E. performed the bioinformatics analysis; J.M.M. and J.B.B. performed the modeling; A.A., E.B., J.M.M., and A.A.J. wrote the paper draft; V.M.G. prepared the figure artwork. All authors contributed to the final version of the paper.

## Competing interests
V.M.G. and E.B. have an equity interest in Synbal, Inc. and Agragene, Inc. companies that may potentially benefit from the research results, and serve on both companies' Scientific Advisory Board and on the Board of Directors of Synbal, Inc. The terms of this arrangement have been reviewed and approved by the University of California, San Diego in accordance with its conflict-of-interest policies. Remaining authors declare no competing interests.
