## [Peer Review File · Nature Communications]

Reviewers' Comments:

Reviewer #1:

Remarks to the Author:

The manuscript of Adolphi et al reports on a second-generation gene drive module that the authors have called Rec (for recoded) that results in the efficient population modification of *An. stephensi* population. This is solid work, a bit incremental but building clearly on previous strains developed from this research group and on a good idea. The manuscript is well written, my overall impression of the manuscript is positive, and I believe publication in *Nat Coms* is merited. I am somewhat puzzled by the lack of exploration on the relevant resistant alleles which I will discuss below. My expertise in gene drive development does not include extensive modelling experience per se – so I will not comment on the quality or methods of that component of the work – a relevant expert in the field of gene drive modelling would be better suited for that.

Major Comments:

As the authors have stated in the manuscript, one of the main challenges of gene drive development is the generation and potential selection of gene drive resistance alleles. The dynamics of selection and the impact of gene drive resistant alleles on gene drive invasiveness depend on a number of factors, most notably the sequence targeted and the effect of its mutation on organismal fitness. The recoded rescue system that is reported here has some interesting implications in resistance allele formation and dynamics.

It is unfortunate that the information regarding the truly interesting type of resistance (namely that which maintains target gene function but defends against nuclease activity) is hardly addressed and feels almost buried in the supplementary information. This data is poorly explained both in the sense of what was done experimentally, why, and what the numbers quoted really are. In the results sections covering the outcome of caged releases (Line 116) the issue of functional alleles appears almost dismissed as irrelevant – which is really frustrating because this issue is now nearly-accepted as the potential demise of nuclease-based gene drives, unless improved engineering solutions – perhaps like those here – are developed.

Undoubtedly, there is something weird going on here, since such resistance alleles would be expected to occur (at least in my opinion) at higher frequencies. The data for example from cage 1:3b suggests that alleles like this are formed and can increase in frequency (6 fold) in the space of a few generations. The authors should expand on this issue in their results (not in the supplement) and rephrase “found no accumulation of specific mutated resistant alleles” as it is currently stated – as this is inconsistent with the results mentioned in the supplement Line 91. Nevertheless, based on the overall data, I conclude that the observed resistant alleles that are functional were not strongly selected (positively) in the presence of the gene drive in the population. The answer to the question “how come” is still pretty unclear to me. The authors argue that a (as in one) $kh+R$ allele tested was no more fit than $khRec+$ and that therefore $khRec+$ outnumbers $kh+R$, due to its biased transmission. Practically that is possible, in the laboratory environment, but how realistic is this in the wild. A related question evolves from this: what would the relative selective advantage be between these two types of alleles, when, to this mix are also thrown anti-parasite effector transgenes. The experimental evaluation of the fitness of $kh+R$ in comparison to $khRec+$ is very strange: a three generation, 1:1 tsunkatse (Sup Table 11), whose results are strange even in G1 (low percentage of $kh+R$) and whose experimental design I cannot find. In comparison, the results and data evaluating fitness of $khRec+$ relative to $kh+$, $kh-$ are very clear, informative and detailed.

A large paragraph is dedicated in the results section on $kh-R$ alleles and how they are removed by mutagenesis of paternal alleles from maternally loaded CRISPR. While this is nice, and it's also nice that the model and experimental data validate the increased removal of $kh-R$ alleles from the population, their presence is actually irrelevant for gene drive unlike what is stated in Line 186.

Perhaps even, early zygotic activity could increase the chance of generating kh+R alleles. So, the fundamental question remains, why were these alleles not taking over the cage population? Perhaps it could be that the sgRNA targets a specific sequence in kh, which when folded into protein, does not accommodate any deviation from the consensus while maintaining function. If this were true, comparative genomic data from other anopheles species could reveal that – low variation at the target site. A second possibility could have to do with the fitness cost of kh mutants and perhaps a nuanced impact on relative fitness to gene drive Rec alleles.

In summary, I think the authors have to address the elephant in the room.

Minor comments

Line 89: how is transmission and conversion defined. I couldn't connect these number and was slightly confused since in the figure the HDR conversion rate is used.

Line 216-20 The logic of this paragraph is a little stretched. The potential fitness cost of an antimalarial effector is unlikely to be mostly dependent on having a longer construct.

Line 221: There is no need to try and market separately SWAP. There is nothing novel here. Call it what it has been called for the last 10 years – cassette exchange.

Reviewer #2:

Remarks to the Author:

The authors describe a modification to their previous attempt to generate a population replacement gene drive approach in the Asian malaria mosquito *An stephensi*, which exhibited a high load do to the cost associated to the integration site into the kh gene which impeded the drive to replace the population. In this study the authors overcome the fitness cost by replacing the anti-plasmodium effectors with a recoded kh sequence that restore a functional copy of the gene. In doing so, the drive relieves the fitness cost do the its integration site, and it can invade efficiently a target population due to its strong super-Mendelian inheritance from three release frequencies. Since the drive allele does not impose a load, both functional and non-functional mutations at the target site due to NHEJ are not favuored over the drive, and do not accumulate over time. The authors use a very nice and clean approach to modify a pre-existing gene drive strain to include the recoded variant using a so-called Swap system. In doing so they also included an attP docking site for subsequent phi31 mediated integration. Unfortunately, this latter information is not clearly stated anywhere in the text. If the authors plan to integrate via the attP a subsequent anti-plasmodium effector, it will also recombine the whole plasmid carrying it, potentially leading to a very large gene drive cassette, impacting the efficacy of HDR, and carrying also unnecessary and potentially harmful sequence (plasmid backbone, antibiotic resistance genes, origin of replication... maybe not harmful per se, for sure from a regulatory point of view in light of field applications). It would have been wiser to introduce two attP sites for recombination-mediated cassette exchange. Of course, Swap system (or similar) can be used to add a cargo, but then I do not see the point of the attP site. More importantly, it is not clear to me why they decided to remove the anti-plasmodium effector generating an effective gene drive but without any functional role. It would have been more useful to include the single chain antibody as well. Or a more effective anti-plasmodium cargo. One reason for doing this is probably to avoid the addition of two fluorescent markers (since the 3xP3-RFP cassette is between the gene drive components and the m2A10 unit that would have been difficult to replace). This should be made clear in the text. The authors would like to assess eventually others anti-parasite effectors once the gene drive is proven to work, but this should be expanded more than the single sentence "our next step includes..." (line 229). The authors should make clear also the distinction between drive transmission and gene conversion (line 89-91).

The population experiments are very clear and well-executed. However, the initial lag of

population 1:3b needs further discussion and the authors cannot claim it is in line 'with analogous cage studies conducted on the nRec drive' (line 147) or that the model fit well the experimental data (line 167-169). In my view, 1:3b dynamics are not captured by the model (see fig 5 and suppl fig 6b). The authors acknowledged that further analysis of those trajectories is required but they could speculate on the reason for such drive lag. One reason for this could be the 'small' population size (200) which is more prone to sampling effect or just stochastic effect when the transgene is at very low frequency, although you should expect this effect to be more prominent in the 1:9 cages, while the three replicates are very reproducible and follow the model. Otherwise, there could be hidden fitness costs associated to the drive that were not identified by fertility assays.

The analysis of resistant alleles provides interesting and not obvious considerations. Non-functional alleles (B) are quickly lost due to negative selection. However, one would expect that a small but consistent fraction of functional resistant alleles (GFP-/Black, R) would constantly be generated at each generation (at low frequency), but those should not be depleted by the drive (as stated in the legend of fig suppl 6, line 72-73) since they are as fit as the H or W allele (or potentially more fit than the H allele), so they should (slowly) accumulate and reach an equilibrium with the H allele, preventing to reach complete fixation (depending on the rate at which they are generated). This is predicted by the model, at least for the 1:3 and 1:9 release (since the drive takes longer to spread, therefore there are more opportunities for the R allele to arise), but is not observed experimentally. Can the authors discuss this? The topic of resistance is one of the most important in gene drive research, especially thinking of field applications. This is not even mentioned in the manuscript. In a larger population, with smaller initial release and a bigger selection pressure, any small difference in fitness have huge implications on the ability of a drive allele to spread and reach fixation. The probability of R alleles to arise is also related to how well conserved the target site is, another consideration missing in the manuscript. Since such drives do not suppress the population, the competition between H and R alleles in the long term is likely to impact the ability of the drive to be effective, especially if the anti-plasmodium effectors are not neutral (and impose any load, even small). This latter consideration is important to be addressed.

This leads me to the main concern I have with this manuscript. Although the authors managed to recover the sterile phenotype of the initial drive by recoding the kh gene in a very elegant way, I do not think this study provides a significant advance in the field. The study provides compelling evidence that a replacement drive can be generated even targeting a non-neutral locus by recoding a functional copy of the target gene. I think the authors missed the opportunity to develop a fully functional gene drive for population replacement which carry a proven anti-malaria effector than can be moved forward in the testing towards field applications. To me this still seems a proof-of-principle demonstration of very nice molecular tools.

My conclusion is that this work is definitely very interesting and relevant to the gene drive field, well executed and ready to be published (after addressing the comments above) but I am not sure it fits to Nature Communication in terms of novelty and advancement in the field.

Minor comment

Fig 1. Panel c should also include the gene drive component under the HDR branch. It is clear to an expert reader the HDR also copies the gene drive unit but it may not be clear to a less experienced reader and it seems only the recoded kh and GFP cassette is copied.

Authors' responses to the Reviewers' comments in bold.

Reviewer #1:

The manuscript of Adolphi et al reports on a second-generation gene drive module that the authors have called Rec (for recoded) that results in the efficient population modification of *An. stephensi* population. This is solid work, a bit incremental but building clearly on previous strains developed from this research group and on a good idea. The manuscript is well written, my overall impression of the manuscript is positive, and I believe publication in Nat Coms is merited. I am somewhat puzzled by the lack of exploration on the relevant resistant alleles which I will discuss below. My expertise in gene drive development does not include extensive modelling experience per se – so I will not comment on the quality or methods of that component of the work – a relevant expert in the field of gene drive modelling would be better suited for that.

Major Comments:

As the authors have stated in the manuscript, one of the main challenges of gene drive development is the generation and potential selection of gene drive resistance alleles. The dynamics of selection and the impact of gene drive resistant alleles on gene drive invasiveness depend on a number of factors, most notably the sequence targeted and the effect of its mutation on organismal fitness. The recoded rescue system that is reported here has some interesting implications in resistance allele formation and dynamics.

It is unfortunate that the information regarding the truly interesting type of resistance (namely that which maintains target gene function but defends against nuclease activity) is hardly addressed and feels almost buried in the supplementary information. This data is poorly explained both in the sense of what was done experimentally, why, and what the numbers quoted really are. In the results sections covering the outcome of caged releases (Line 116) the issue of functional alleles appears almost dismissed as irrelevant – which is really frustrating because this issue is now nearly-accepted as the potential demise of nuclease-based gene drives, unless improved engineering solutions – perhaps like those here – are developed.

We agree with the reviewer on the importance of this topic and have moved the data on the creation of resistant functional alleles from the Supplementary Information to the Results under a dedicated heading (lines 197-239). In addition, the cage experiments in which a functional drive-resistant EJ allele (kh^{+R}) competes against the drive allele (kh^{Rec+}) have now been carried out for 6 generations and are reported in Fig. 5 (previously Supplementary Figure 11). These experiments provide strong support for the hypothesis that kh^{+R} , which encodes a modified functional version of the KH protein, gradually disappears over multiple generations. Thus, this allele is certainly not more fit than the kh^{Rec+} drive allele, which encodes a fully-native protein. Details on the experimental design used to assess the presence of resistant alleles and the dedicated cage experiments conducted to assess drive dynamics in the presence of a functional resistant allele also have been expanded in the Methods (lines 551-559) and in the Discussion (lines 313-341) to reflect the importance of this finding and the ability of this drive to overcome functional resistance. This is discussed in more details in a related comment below.

Undoubtedly, there is something weird going on here, since such resistance alleles would be expected to occur (at least in my opinion) at higher frequencies. The data for example from cage 1:3b suggests that alleles like this are formed and can increase in frequency (6 fold) in the space of a few generations. The authors should expand on this issue in their results (not in the supplement) and rephrase "found no accumulation of specific mutated resistant alleles" as it is currently stated – as this is inconsistent with the results mentioned in the supplement Line 91.

Our sequencing data and modeling are in line with the creation of functional, potentially drive-resistant alleles at low rates. Also, we note that following the appearance and initial increase in frequency of the functional, potentially drive-resistant allele kh^{+R} , its frequency stabilized and did not increase above 6% from generation 11 through generation 20. This result provides another strong line of support for our hypothesis that functional EJs, when they do arise, do not take over the population since they are not more fit than the drive element. As Fig. 5 shows, direct competition between kh^{+R} and the kh^{Rec+} drive resulted in the latter slowly displacing the functional EJ variant. The situation in Cage 1:3_B may be the same, although the frequency of the EJ allele never got high enough to make this comparison. We justify this in a new paragraph inserted in the Discussion regarding functional constraints of the targeted locus (as also suggested by this reviewer in a later comment) (lines 313-341).

We modified the text to read ‘we observed only a modest accumulation of this mutation amongst the non-drive alleles that did not prevent drive invasion’ (lines 331-332) as indeed we observed an increase in the functional resistance allele kh^{+R} frequency from 1% to 12% in the span of 6 generations. We refer to it as a ‘modest increase’ in the context of the drive allele concomitantly increasing in frequency from 53% to 97%.

Overall, while functional resistant alleles are formed rarely, they do not prevent the nine cages tested from reaching maximum introduction of the drive.

Nevertheless, based on the overall data, I conclude that the observed resistant alleles that are functional were not strongly selected (positively) in the presence of the gene drive in the population. The answer to the question “how come” is still pretty unclear to me. The authors argue that a (as in one) kh^{+R} allele tested was no more fit than kh^{Rec+} and that therefore kh^{Rec+} outnumbers kh^{+R} , due to its biased transmission. Practically that is possible, in the laboratory environment, but how realistic is this in the wild. A related question evolves from this: what would the relative selective advantage be between these two types of alleles, when, to this mix are also thrown anti-parasite effector transgenes. The experimental evaluation of the fitness of kh^{+R} in comparison to kh^{Rec+} is very strange: a three generation, 1:1 tsunkatse (Sup Table 11), whose results are strange even in G1 (low percentage of kh^{+R}) and whose experimental design I cannot find. In comparison, the results and data evaluating fitness of kh^{Rec+} relative to kh^{+} , kh^{-} are very clear, informative and detailed.

We propose a scenario where the functional resistant allele kh^{+R} is either neutral or less fit relative to the drive allele, and this is consistent with the observation that this allele does not outcompete the drive.

Here, we refer to ‘allele fitness’ in terms of how competitive the functional, potentially drive-resistant allele is in the presence of the drive allele when starting at the same frequency in cage experiments. These are done by setting up cages where individuals heterozygous for a copy of the drive and a copy of the resistant functional allele (kh^{Rec+}/kh^{+R}) can mate and observing the relative frequency of the associated eye markers to infer relative allele frequencies (new Fig. 5). In such experiments, which have now been carried for an additional 3 generations (total of 6 generations), equal competitiveness of the two alleles would result in a sustained phenotype ratio of 3:1 $GFP^{+}:GFP^{-}$, while deviations from this ratio are indicative of unequal fitness (due to any number of factors). We observed that the drive allele outcompetes the functional resistant allele, most likely due to a fitness load associated with the mutation present in this allele in a site that is functionally-constrained as addressed in more detail in our next comment. These points are addressed in the text in the following sections: Results (lines 221-239), Discussion (lines 313-341), and Methods (lines 544-559).

The reviewer is correct to note that experiments conducted in the laboratory may not reflect outcomes in the wild. However, cage experiments are informative in assessing important factors influencing competitive outcomes. The overall findings in this and prior studies strongly support the hypothesis that recoded modification gene-drives powered by lethal/sterile mosaicism should reach high levels of introduction into populations. How long they will last will depend, as the reviewer notes, on many secondary factors that will need to be assessed in field studies, which we hope the current study will help motivate.

We added a paragraph in the Discussion (lines 342-356) to address the comment on possible fitness loads associated with the presence of effector molecules. The effector molecules we aim to use, namely single chain antibody fragments, have been shown to be associated with little if any fitness load in the mosquito, probably due to their high specificity for *Plasmodium* epitopes. While this will have to be ascertained empirically in the final mosquito strain carrying both the drive and the effectors, we do not expect a major impact of these molecules on mosquito fitness.

A large paragraph is dedicated in the results section on kh-R alleles and how they are removed by mutagenesis of paternal alleles from maternally loaded CRISPR. While this is nice, and it's also nice that the model and experimental data validate the increased removal of kh-R alleles from the population, their presence is actually irrelevant for gene drive unlike what is stated in Line 186. Perhaps even, early zygotic activity could increase the chance of generating kh+R alleles. So, the fundamental question remains, why were these alleles not taking over the cage population? Perhaps it could be that the sgRNA targets a specific sequence in kh, which when folded into protein, does not accommodate any deviation from the consensus while maintaining function. If this were true, comparative genomic data from other anopheles species could reveal that – low variation at the target site. A second possibility could have to do with the fitness cost of kh mutants and perhaps a nuanced impact on relative fitness to gene drive Rec alleles.

As stated above, our data support the conclusion that the functional mutations present in the kh^{+R} allele are not sufficient to restore fully wild-type levels of KH enzyme, unlike the drive element, which restores wild-type levels of KH enzyme. Therefore, the resistant functional allele is not positively-selected relative to the drive as demonstrated in allele competition experiments. In further support of this conclusion, we introduced a paragraph (lines 313-341) addressing the high levels of conservation amongst insects and in higher organisms in the amino acid composition of the KH enzyme at the site of cleavage. The site targeted by our gRNA is associated with strong functional constraints, which, along with evidence from a previous cage experiment using the same gRNA (Pham *et al.*, 2019), supports the hypothesis that mutations at this site are not (or rarely) tolerated.

We note that in separate studies examining the effect of drive-neutralizing elements, we have targeted the Cas9 protein for mutagenesis with gRNAs directed at targets encoding amino acid residues in catalytically-essential regions of the protein. We tested two such gRNAs and, while both of them generated in-frame indels ~33% of the time (as might be expected on average), over 99% of such alleles were non-functional as assessed by several measures of Cas9 activity in both single-generation crosses and in multigenerational cage studies. Thus, the strategy of inserting a gene-drive into a functionally-essential site in a gene providing a recoded sequence to restore native function of that gene, and lethal/sterile mosaicism to dominantly eliminate the vast majority of non-functional EJ alleles is broadly generalizable. Indeed, in yet unpublished experiments in fruit flies, we have generated five highly-efficient recoded drive systems that similarly depend on lethal or sterile mosaicism and target both sexes to provide rapid invasive drive dynamics.

In summary, I think the authors have to address the elephant in the room.

We hope that the reviewer agrees that we have addressed the functional EJ question in a satisfactory fashion.

Minor comments

Line 89: how is transmission and conversion defined. I couldn't connect these number and was slightly confused since in the figure the HDR conversion rate is used.

Definitions of drive transmission and gene conversion (or HDR rate) were added in the Results (lines 111-114) and in the Methods (lines 443-446). All raw data on primary drive transmission and HDR rate calculations reported in the text and in Fig. 2 have been added to Supplementary Table 2. A note has been added in the legend of Fig. 2 to clarify how we calculate the various indices of drive (lines 750-751). Furthermore, the text has been modified to accommodate new more comprehensive data on the primary transmission of *Reckh* through the female and the male lineages obtained recently from replicate sets of cages.

Line 216-20 The logic of this paragraph is a little stretched. The potential fitness cost of an antimalarial effector is unlikely to be mostly dependent on having a longer construct.

This paragraph has been expanded to address the potential issue of fitness costs associated with the antimalarial effectors (lines 342-356). Also, we note and reference prior work indicating that mosquitoes carrying two different anti-malarial effectors displayed no obvious fitness costs in the laboratory when tested extensively by several measures including mating efficiency, egg laying, and longevity.

Line 221: There is no need to try and market separately SWAP. There is nothing novel here. Call it what it has been called for the last 10 years – cassette exchange.

We refer to Swap throughout the text as ‘Cas9/gRNA-mediated cassette exchange’. Indeed, while Swap results in the exchange of a previously-integrated cassette, the molecular bases of these processes are different from those underlying the $\phi C31$ recombinase-mediated cassette-exchange (RMCE). Swap is not a process driven by recombination of docking sites but relies on the concerted action of two gRNAs and Cas9. The relative advantages of Swap over RMCE, namely the independence from preinserted *attP* docking sites, and its possible applications are discussed in lines 357-374. While such CRISPR-mediated cassette exchange has been demonstrated in fruit flies (we have led in developing such tools), to our knowledge, this is the first time that this efficient transgenesis strategy has been applied to mosquitoes. Since transgenesis in mosquitoes poses a much larger technical hurdle than in other organisms, this represents a significant technical advance that others in the field will hopefully be encouraged to try based on our success.

Reviewer #2 (Remarks to the Author):

The authors describe a modification to their previous attempt to generate a population replacement gene drive approach in the Asian malaria mosquito *An stephensi*, which exhibited a high load due to the cost associated to the integration site into the *kh* gene which impeded the drive to replace the population. In this study the authors overcome the fitness cost by replacing the anti-plasmodium effectors with a recoded *kh* sequence that restores a functional copy of the gene. In doing so, the drive relieves the fitness cost due to its integration site, and it can invade efficiently a target population due to its strong super-Mendelian inheritance from three release frequencies. Since the drive allele does not impose a load, both functional and non-functional mutations at the target site due to NHEJ are not favored over the drive, and do not accumulate over time.

The authors use a very nice and clean approach to modify a pre-existing gene drive strain to include the recoded variant using a so-called Swap system. In doing so they also included an *attP* docking site for subsequent $\phi 31$ mediated integration. Unfortunately, this latter information is not clearly stated anywhere in the text. If the authors plan to integrate via the *attP* a subsequent anti-plasmodium effector, it will also recombine the whole plasmid carrying it, potentially leading to a very large gene drive cassette, impacting the efficacy of HDR, and carrying also unnecessary and potentially harmful sequence (plasmid backbone, antibiotic resistance genes, origin of replication... maybe not harmful per se, for sure from a regulatory point of view in light of field applications). It would have been wiser to introduce two *attP* sites for recombination-mediated cassette exchange. Of course, Swap system (or similar) can be used to add a cargo, but then I do not see the point of the *attP* site.

We share with the reviewer an understandable concern regarding the potential negative effect of the integration via the *attP* site of the whole donor plasmid. Indeed, we do not plan to introduce antimalarial effector molecules via this site but rather by using Swap or by building and injecting a new construct altogether. Nevertheless, the *Reckh* line can be used as a standard docking line as well to test the blocking capabilities of different combinations of effector

molecules when expressed in the genomic environment of the *kh* locus and independent of the drive process. This point is addressed in the discussion (lines 367-371).

More importantly, it is not clear to me why they decided to remove the anti-plasmodium effector generating an effective gene drive but without any functional role. It would have been more useful to include the single chain antibody as well. Or a more effective anti-plasmodium cargo. One reason for doing this is probably to avoid the addition of two fluorescent markers (since the 3xP3-RFP cassette is between the gene drive components and the m2A10 unit that would have been difficult to replace). This should be made clear in the text.

As the reviewer suggests, the antimalarial effector molecules present in the original nRec construct were removed to accommodate partial recoding of the *kh* gene while avoiding the presence of a double fluorescent marker. This point is now clarified in the Discussion (lines 344-346). Additionally, we wished to apply a phased approach by testing 1) the drive capabilities in the absence of confounding factors and 2) the suitability of the *kh* locus to efficiently sustain the recoding strategy.

The authors would like to assess eventually others anti-parasite effectors once the gene drive is proven to work, but this should be expanded more than the single sentence “our next step includes...” (line 229).

An expanded consideration of our strategy for population modification based on single chain antibody fragments as antimalarial molecules, including the predicted absence of major fitness cost associated to them, is now included in the Discussion (lines 342-353).

The authors should make clear also the distinction between drive transmission and gene conversion (line 89-91).

As addressed in a similar comment from Reviewer 1, definition of drive transmission and gene conversion (or HDR rate) were added in the Results (lines 111-114) and in the Methods (lines 443-446). These two measures are also indicated separately in Fig. 2 to clarify this important difference.

The population experiments are very clear and well-executed. However, the initial lag of population 1:3b needs further discussion and the authors cannot claim it is in line ‘with analogous cage studies conducted on the nRec drive’ (line 147) or that the model fit well the experimental data (line 167-169). In my view, 1:3b dynamics are not captured by the model (see fig 5 and suppl fig 6b). The authors acknowledged that further analysis of those trajectories is required but they could speculate on the reason for such drive lag. One reason for this could be the ‘small’ population size (200) which is more prone to sampling effect or just stochastic effect when the transgene is at very low frequency, although you should expect this effect to be more prominent in the 1:9 cages, while the three replicates are very reproducible and follow the model. Otherwise, there could be hidden fitness costs associated to the drive that were not identified by fertility assays.

Regarding the delay in drive invasion observed in one of the cages, the text has been modified to include the sentence ‘it remains difficult to explain since it is not fully supported by the predictions of the stochastic model’ (lines 306-307). As the reviewer suggests, we do not expect this to result from a sampling effect as this would have been observed in other cages (specifically the ones seeded with a lower release ratio). Also, it is highly unlikely that the sampling effect would justify a drive lag for multiple consecutive generations. Our sequencing analysis also excludes the involvement of EJ-induced mutations at the target site, including those that restore gene function, from causing it. The underlying causes of this lag deserves additional investigation and would depend on being able to reproduce it consistently, but these experiments are beyond the scope of this paper. These points have been raised in the Discussion (lines 305-312).

The analysis of resistant alleles provides interesting and not obvious considerations. Non-functional alleles (B) are quickly lost due to negative selection. However, one would expect that a small but consistent fraction of functional resistant alleles (GFP-/Black, R) would constantly be generated at each generation (at low frequency), but those should

not be depleted by the drive (as stated in the legend of fig suppl 6, line 72-73) since they are as fit as the H or W allele (or potentially more fit than the H allele), so they should (slowly) accumulate and reach an equilibrium with the H allele, preventing to reach complete fixation (depending on the rate at which they are generated). This is predicted by the model, at least for the 1:3 and 1:9 release (since the drive takes longer to spread, therefore there are more opportunities for the R allele to arise), but is not observed experimentally. Can the authors discuss this?

The topic of resistance is one of the most important in gene drive research, especially thinking of field applications. This is not even mentioned in the manuscript.

Discussion of the generation and impact of resistance alleles on drive dynamics has been expanded in the text throughout the manuscript including the Results section (the data presented in the supplementary text has been moved into the main text with a dedicated heading - lines 196-239 - and main Fig. 5), and in the Discussion (lines 313-341) as detailed in the next comment. In brief, we believe that functional resistant alleles are only generated rarely, and they form in proportion to the frequency of the drive in the population. So, they do accumulate, but do so in a constant proportion so that their final levels are essentially the same as the rate at which they are generated, which is quite low. Crucially, in the cages where they are formed, they do not outcompete the kh^{Rec+} drive. In fact, in a now extended 1:1 competition between such a drive-resistant functional allele (kh^{+R}) and the kh^{Rec+} drive, the drive slowly takes over. Thus, we believe that if the gRNA is well-chosen to target a region of the gene essential to its function (e.g. a catalytic center) then the probability of such alleles forming will be low. Indeed, as indicated above in response to Reviewer 1, in separate studies we have designed two gRNAs targeting the Cas9 enzyme in critical sites and virtually never recovered functional INDELS despite a third of these events being in-frame. These important points on functional constraints at the cut site have been added to the revised Discussion (lines 313-341).

Overall, the topic of resistance has been expanded throughout the text in the Introduction (lines 44-52), Result (lines 221-239), and Discussion (lines 313-341) sections.

In a larger population, with smaller initial release and a bigger selection pressure, any small difference in fitness have huge implications on the ability of a drive allele to spread and reach fixation. The probability of R alleles to arise is also related to how well conserved the target site is, another consideration missing in the manuscript. Since such drives do not suppress the population, the competition between H and R alleles in the long term is likely to impact the ability of the drive to be effective, especially if the anti-plasmodium effectors are not neutral (and impose any load, even small). This latter consideration is important to be addressed.

The reviewer's comment on the accumulation of functional resistant (R) alleles assumes equal competitiveness (or fitness) amongst alleles that restore the WT black-eye phenotype, namely the drive and the functional resistant alleles. However, in this study the drive allele restores the endogenous WT kh sequence while the R allele observed (kh^{+R}) causes a non-synonymous mutation. To discuss this in more detail, we included, as suggested, a paragraph on the conservation of the kh locus and its strong functional constraints (lines 313-341). The trajectories observed in the cage experiments are consistent with 1) the fact that functional resistance alleles are rarely created in the first place (as also supported by the model) and 2) that even when functional mutations are created, these are unlikely to sustain WT levels of protein function due to the strong functional constraints of the coding sequence at the cut site. Indeed, we recovered in-frame mutations in individuals with white eyes. Overall, we found no evidence in our experiments of positive selection of non-drive functional alleles (Cas9-derived or potentially already present in our colony cage) or evidence of these alleles interfering with the drive reaching maximum prevalence.

The potential impact of the effector molecules on mosquito fitness is included in response to a similar comment from Reviewer 1 and is discussed in lines 342-353. As indicated above, prior experiments have examined potential fitness costs of the anti-malarial effectors in a laboratory setting and did not find any. However, the reviewer is correct in expressing concerns about this issue in the context of natural populations. Even if it were the case that there are fitness costs associated with the drive element, the issue will be 'does the drive hold long enough to help reduce

parasite transmission to the point where it disappears locally?'. The only way to know whether this will prove to be a major concern is to conduct field trials, which we hope this study will help justify.

This leads me to the main concern I have with this manuscript. Although the authors managed to recover the sterile phenotype of the initial drive by recoding the kh gene in a very elegant way, I do not think this study provides a significant advance in the field. The study provides compelling evidence that a replacement drive can be generated even targeting a non-neutral locus by recoding a functional copy of the target gene.

The goal of the Rec system was not only to demonstrate the ability of recovering the lethal/sterile phenotype at the target locus by recoding it, but in doing so this creates the conditions for the active elimination of resistant non-functional alleles and, as a result of a combination of target-site constraints and the ability of the drive to restore the endogenous sequence of the target gene, reduces the potentially deleterious effect of functional resistant alleles on the drive success. The drive is as effective as it is *because* it targets a non-neutral locus, not *despite of* it. Therefore, with respect, we beg to disagree with the reviewer about the significance of the work. To use the reviewer's words, we have shown 'compelling evidence that a replacement drive can be generated even targeting a non-neutral locus by recoding a functional copy of the target gene.' This IS a significant advance in that it shows how a wider scope of target genes can be used, some of which may contribute effectively to the function of the desired driven traits. Indeed, we have now developed several other recoded gene-drives inserted into loci essential for viability or fertility in both sexes in fruit flies and have observed that all of these systems demonstrate the benefit of lethal or sterile mosaicism driving to nearly full introduction with rapid kinetics. The current study is the first demonstration of this important and generalizable design strategy in mosquitoes and is a game-changing advance for population modification approaches.

I think the authors missed the opportunity to develop a fully functional gene drive for population replacement which carry a proven anti-malaria effector than can be moved forward in the testing towards field applications. To me this still seems a proof-of-principle demonstration of very nice molecular tools.

My conclusion is that this work is definitely very interesting and relevant to the gene drive field, well executed and ready to be published (after addressing the comments above) but I am not sure it fits to Nature Communication in terms of novelty and advancement in the field.

We respectfully disagree with this assessment. As in our previous response, we emphasize that this work provides a large step forward by providing a general and potent solution to the drive-resistance problem that has been suggested as a fatal flaw of CRISPR based drive systems. By providing a generally-applicable strategy to defeat this problem, we demonstrate that such concerns may not present a serious challenge to population modification strategies as part of a phased approach for developing technologies to aid in the global malaria eradication agenda. We can now turn our attention to optimizing antiparasitic effector molecules based on our confidence that the drive components will be effective in achieving high levels of introduction in mosquito populations.

Minor comment

Fig 1. Panel c should also include the gene drive component under the HDR branch. It is clear to an expert reader the HDR also copies the gene drive unit but it may not be clear to a less experienced reader and it seems only the recoded kh and GFP cassette is copied.

We have followed this helpful suggestion.

Reviewers' Comments:

Reviewer #1:

Remarks to the Author:

I have now reviewed the revised manuscript of Adolphi et al. I would like to apologise to the authors for the slow return of my review. The manuscript has been significantly improved and publication at Nature Comms should proceed, following a few additional recommendations that I hope the authors will agree with. Before discussing these details, I would like to encourage authors to deposit the manuscript to the bioRxiv before journal submission and not after the first round of review. There is no reason to delay pre-publication spreading of data and information, especially given the number of groups that are actively working on similar designs at the same time.

Regarding the data and analysis of functional resistant alleles, I find the experiments and their description/visibility in the manuscript much improved. Thank you. I do not believe additional experiments or analyses are needed.

Regarding the issue of additional loads imposed by malaria effectors, again the authors have sufficiently expanded and highlighted the issues in the discussion.

Overall comments regarding this design for population replacement that could be addressed and perhaps discussed with more rigor.

1) The authors "sell" this design as a resistance mitigation strategy (line 53 and 64). However, I wonder how much this is making lemonade out of lemons – the lemons being the high rate of activity from maternally loaded Cas9/sgRNAs leading to the formation of resistance alleles in the female line. As the authors have themselves highlighted in the discussion (line 384), substituting the current vasa promoter for the more tightly regulated nanos or zpg promoters could improve the performance of the drive. The question is whether using these promoters would remove the maternal deposition and thus reduce resistance allele formation through end join which leads to the reduction of homing rates, in daughters of drive females. Since the targeted region appears to be constrained in the production of resistance functional alleles, perhaps this would have been a better design strategy – in that it is simpler and does not require starting off with a female-specific essential gene.

2) While the authors have improved the discussion regarding the importance of resistant-functional alleles and shown that these are most likely neutral (compared to the drive allele), they have not discussed what would happen in context of population replacement – namely how would malaria transmission be affected if a significant percentage of the population carries drive-resistant alleles. The results from the 1:3 cages suggest that the generation and maintenance of such alleles is pretty stochastic. And of course, there is the compounded issue of changes in gene drive behavior and cost through both the inclusion of anti-malarial effectors, but also in the wild, the cost or lack of, Plasmodium infection. The manuscript modelling describes well the dynamics of the drive in the population and the impact that functional resistance alleles have, but the discussion and modelling misses an opportunity to frame these within the context of population replacement.

Minor comments:

Line 46-48: resistant alleles can also be generated in the germline

Line 64: This system eliminates non-functional EJ events by eliminating individuals that carry them, similarly to CLVR.

Line 91: Missing the point: it is not just that the homozygous or heteroallelic LOF kh alleles impair female survival that results in limiting the spread, it is maternal deposition that does this (as originally proposed by Papathanos et al 2008, 12 years ago).

Line 194: It is unclear from the description whether the nRec alleles went to fixation which in turn led to extinction.

Yours sincerely
Phi Papathanos

Reviewer #2:

Remarks to the Author:

The authors largely improved the manuscript taking into account all the concerns raised by the reviewers. The additional data provided and the more detailed discussion make the manuscript clearer, more balanced and more accurate. I thank the authors for improving the manuscript which I think is now suitable for publication.

Minor comments:

Line 193: reached >95% introduction before driving to extinction due to fixation of kh double-mutant genotypes and associated female load.

It is not clear what is driving to extinction. As it is written, it seems the cages is the subject.

Line 377. I would be more cautious to state that 'the characteristics of the Reckh gene-drive conform with those defined as part of a proposed TPP', as it is, since it is missing the anti-malaria effector. I would include 'once the addition of the anti-malaria effectors are proven effective under this configuration' (or similar). It is true that the authors predict that antimalaria effectors are unlikely to affect drive (line 353-354), and I might agree. However, to my knowledge this is still to be proven.

NCOMMS-20-20211: Efficient population modification gene-drive rescue system in the malaria mosquito *Anopheles stephensi*' by Adriana Adolphi, Valentino M. Gantz, Nijole Jasinskiene, Hsu-Feng Lee, Kristy Hwang, Emily A. Bulger, Arunachalam Ramaiah, Jared B. Bennett, Gerard Terradas, J.J. Emerson, John M. Marshall, Ethan Bier and Anthony A. James.

Authors' (in bold) responses to Reviewers' comments. Line numbers refer to the marked (tracked changes visible) version of the manuscript.

We thank the two reviewers for their time and thoughtful comments that have improved the final version of the manuscript.

Reviewer #1:

I have now reviewed the revised manuscript of Adolphi et al. I would like to apologise to the authors for the slow return of my review. The manuscript has been significantly improved and publication at Nature Comms should proceed, following a few additional recommendations that I hope the authors will agree with. Before discussing these details, I would like to encourage authors to deposit the manuscript to the bioRxiv before journal submission and not after the first round of review. There is no reason to delay pre-publication spreading of data and information, especially given the number of groups that are actively working on similar designs at the same time.

Regarding the data and analysis of functional resistant alleles, I find the experiments and their description/visibility in the manuscript much improved. Thank you. I do not believe additional experiments or analyses are needed.

Regarding the issue of additional loads imposed by malaria effectors, again the authors have sufficiently expanded and highlighted the issues in the discussion.

Overall comments regarding this design for population replacement that could be addressed and perhaps discussed with more rigor.

1) The authors “sell” this design as a resistance mitigation strategy (line 53 and 64). However, I wonder how much this is making lemonade out of lemons – the lemons being the high rate of activity from maternally loaded Cas9/sgRNAs leading to the formation of resistance alleles in the female line. As the authors have themselves highlighted in the discussion (line 384), substituting the current vasa promoter for the more tightly regulated nanos or zpg promoters could improve the performance of the drive. The question is whether using these promoters would remove the maternal deposition and thus reduce resistance allele formation through end join which leads to the reduction of homing rates, in daughters of drive females. Since the targeted region appears to be constrained in the production of resistance functional alleles, perhaps this would have been a better design strategy – in that it is simpler and does not require starting off with a female-specific essential gene.

We provide empirical evidence that this *is* indeed a mitigation strategy. At the time that the original non-recoded line was developed (2015), no one knew what the empirical outcomes would be. We followed up on this with extensive small cage trials (2019). We then asked whether we could correct this prototype while taking advantage of targeting a haplosufficient locus. Here we provide the first evidence of a rescue system in mosquitoes. A few rescue systems have now been proposed but have only been tested in the fruit fly, *Drosophila melanogaster*.

What we now know does foster different, improved designs. We have discussed that the system may be rendered even more efficient by 1) the use of more tightly regulated promoters (lines 398-401) or by targeting genes that are essential for viability and or fertility in both sexes (lines 328-329). However, that

does not take away from what was done here, demonstrating an effective retrofit on a powerful drive system.

Importantly, we believe that lethal-mosaicism provides a general solution to the problem of females generating EJ alleles and should allow for the design of efficient gene drives at many different target loci and insect species where identification of ‘ideal’ promoters may not be straightforward or transferable (lines 384-387). For example, even within *Anopheles* mosquitoes, it is not clear what the *An. stephensi nanos* regulatory sequences are that correspond to those used successfully in *An. gambiae*. The Bier laboratories’ recent work in fruit flies supports the conclusion that this strategy is indeed generalizable to other essential loci as they have developed recoded drives in several essential genes that display highly-efficient transmission and drive.

2) While the authors have improved the discussion regarding the importance of resistant-functional alleles and shown that these are most likely neutral (compared to the drive allele), they have not discussed what would happen in context of population replacement – namely how would malaria transmission be affected if a significant percentage of the population carries drive-resistant alleles. The results from the 1:3 cages suggest that the generation and maintenance of such alleles is pretty stochastic. And of course, there is the compounded issue of changes in gene drive behavior and cost through both the inclusion of anti-malarial effectors, but also in the wild, the cost or lack of, Plasmodium infection. The manuscript modelling describes well the dynamics of the drive in the population and the impact that functional resistance alleles have, but the discussion and modelling misses an opportunity to frame these within the context of population replacement.

We agree that these topics are important, but extensive comments on them would be speculative even if supported by modelling. We are pursuing empirical studies of population replacement features including the efficacy of the effector molecules, efficacy of the introduction of the drive system in the target wild population, and the impact of a threshold requirement for the parasite inoculum to cause disease in humans. While modeling is useful, these questions all require empirical answers if the population replacement strategy is to be continued to be developed.

Our goal is to develop anti-malarial effector genes whose expression and/or products block parasite transmission as a single genetic copy (heterozygous). For such effectors, the level of population conversion we observe in this study would be predicted to be sufficient to have an appreciable impact on disease transmission. We also are developing drive systems that will lead to lethality or sterility of both sexes, which in combination with more tightly-regulated promoters, should reduce the fraction of resistant alleles to levels low enough that nearly all individuals would eventually be homozygous for the element. We also note that the slow Mendelian selection against the great majority of loss-of-function resistant alleles will gradually cull them from the population. These features can all be incorporated along with modeling results in final target product profiles that aim to achieve a stated level of parasite control.

Minor comments:

Line 46-48: resistant alleles can also be generated in the germline

The text was modified to ‘developmental stages and tissues, including the germline and somatic tissues during early embryogenesis’ [line 54].

Line 64: This system eliminates non-functional EJ events by eliminating individuals that carry them, similarly to CLVR.

Reckh, CleaveR and Toxin-Antidote systems all target essential genes and eliminate non-functional mutations by eliminating individuals that carry them.

Line 91: Missing the point: it is not just that the homozygous or heteroallelic LOF *kh* alleles impair female survival that results in limiting the spread, it is maternal deposition that does this (as originally proposed by Papathanos et al 2008, 12 years ago).

This sentence refers to the non-recoded *kh* gene drive. In both non-recoded and recoded drives lethal/sterile mosaicism derived from maternal deposition is the mechanism for the generation of LOF *kh* alleles, but it is the actual phenotype of these alleles that impairs female survival. This is demonstrated further by the data showing that homozygous insertion of the nRec drive into the target locus contributes substantially to population extinction without invoking maternal deposition.

Line 194: It is unclear from the description whether the nRec alleles went to fixation which in turn led to extinction.

The text was modified to clarify : ‘only two of the three 1:1 and 1:3 nRec cages, and none of the 1:9 replicates reached >95% introduction before either the mosquito population crashed due to fixation of *kh* double-mutant genotypes or the drive was selected out of the population’ [lines 209-211].

Reviewer #2:

The authors largely improved the manuscript taking into account all the concerns raised by the reviewers. The additional data provided and the more detailed discussion make the manuscript clearer, more balanced and more accurate. I thank the authors for improving the manuscript which I think is now suitable for publication.

Minor comments:

Line 193: reached >95% introduction before driving to extinction due to fixation of *kh* double-mutant genotypes and associated female load.

It is not clear what is driving to extinction. As it is written, it seems the cages is the subject.

Following similar comment from both reviewers, the text was modified to ‘only two of the three 1:1 and 1:3 nRec cages, and none of the 1:9 replicates reached >95% introduction before either the mosquito population crashed due to fixation of *kh* double-mutant genotypes or the drive was selected out of the population’ [lines 209-211].

Line 377. I would be more cautious to state that ‘the characteristics of the Reckh gene-drive conform with those defined as part of a proposed TPP’, as it is, since it is missing the anti-malaria effector. I would include ‘once the addition of the anti-malaria effectors are proven effective under this configuration’ (or similar). It is true that the authors predict that antimalaria effectors are unlikely to affect drive (line 353-354), and I might agree. However, to my knowledge this is still to be proven.

The text was modified to ‘the laboratory assessments conducted so far, carried out in line with the recommended phased pathway for testing gene-drive mosquitoes, suggest that the characteristics of the *Reckh* gene-drive are likely to conform with those defined as part of a proposed Target Product Profile for population modification of mosquito strains once the addition of the anti-malaria effectors is proven effective under this configuration’ [lines 405-408].